# HO$_x$ and NO$_x$ production in oxidation flow reactors via photolysis of isopropyl nitrite, isopropyl nitrite-d$_7$, and 1,3-propyl dinitrite at $\lambda$ = 254, 350, and 369 nm

Andrew Lambe[1], Jordan Krechmer[1], Zhe Peng[2], Jason Casar[3], Anthony Carrasquillo[4], Jonathan Raff[5], Jose Jimenez[2], and Douglas Worsnop[1,6]

[1]Center for Aerosol and Cloud Chemistry, Aerodyne Research Inc., Billerica, MA, USA
[2]Dept. of Chemistry and Cooperative Institute for Research in Environmental Sciences (CIRES), University of Colorado, Boulder, CO, USA
[3]Dept. of Chemistry, Harvey Mudd College, Claremont, CA, USA
[4]Dept. of Chemistry, Williams College, Williamstown, MA, USA
[5]School of Public and Environmental Affairs, Indiana University, Bloomington, IN, USA
[6]Dept. of Physics, University of Helsinki, Helsinki, FI

**Correspondence:** Andrew T. Lambe (lambe@aerodyne.com), Zhe Peng (zhe.peng@colorado.edu)

**Abstract.** Oxidation flow reactors (OFRs) are an emerging technique for studying the formation and oxidative aging of organic aerosols and other applications. In these flow reactors, hydroxyl radicals (OH), hydroperoxyl radicals (HO$_2$), and nitric oxide (NO) are typically produced in the following ways: photolysis of ozone (O$_3$) at $\lambda$ = 254 nm, photolysis of H$_2$O at $\lambda$ = 185 nm, and via reactions of O($^1$D) with H$_2$O and nitrous oxide (N$_2$O); O($^1$D) is formed via photolysis of O$_3$ at $\lambda$ = 254 nm and/or N$_2$O at $\lambda$ = 185 nm. Here, we adapt a complementary method that uses alkyl nitrite photolysis as a source of OH via its production of HO$_2$ and NO followed by the reaction NO + HO$_2$ → NO$_2$ + OH. We present experimental and model characterization of the OH exposure and NO$_x$ levels generated via photolysis of C$_3$ alkyl nitrites (isopropyl nitrite, perdeuterated isopropyl nitrite, 1,3-propyl dinitrite) in the Potential Aerosol Mass (PAM) OFR as a function of photolysis wavelength ($\lambda$ = 254 to 369 nm) and organic nitrite concentration (0.5 to 20 ppm). We also apply this technique in conjunction with chemical ionization mass spectrometer measurements of multifunctional oxidation products generated following the exposure of $\alpha$-pinene to HO$_x$ and NO$_x$ obtained using both isopropyl nitrite and O$_3$ + H$_2$O + N$_2$O as the radical precursors.

## 1 Introduction

Hydroxyl (OH) radicals govern the concentrations of most atmospheric organic compounds, including those that lead to secondary organic aerosol (SOA) formation. The relative importance of different primary OH precursors varies in different parts of the atmosphere, and may include contributions from O($^1$D)-H$_2$O reactions, hydrogen peroxide (H$_2$O$_2$), methyl peroxide (CH$_3$OOH), nitrous acid (HONO) photolysis, and ozone-alkene reactions. Additionally, ozone-hydroperoxy (HO$_2$) reactions and NO-HO$_2$ reactions recycle HO$_2$ back to OH (Mao et al., 2009; Lee et al., 2016). For decades, a handful of radical precursors have been used to generate OH radicals in the laboratory to initiate SOA production under controlled conditions. Environmental chambers most commonly photolyze nitrous acid (HONO), methyl nitrite (CH$_3$ONO), or hydrogen peroxide

($H_2O_2$) at $\lambda > 310$ nm to mimic SOA production, over experimental timescales of hours to days, simulating up to 2 days of equivalent atmospheric exposure (Atkinson et al., 1981; Matsunaga and Ziemann, 2010; Chhabra et al., 2011; Finewax et al., 2018).

Oxidation flow reactors (OFRs) photolyze $H_2O$ and $O_3$ at $\lambda = 185$ and 254 nm over experimental timescales of minutes, simulating multiple days of equivalent atmospheric exposure (Lambe et al., 2012; Peng et al., 2015). Recent application of $O(^1D) + H_2O + N_2O$ reactions to study $NO_x$-dependent SOA formation pathways facilitated characterization of oxidation products generated over a range of low- to high-$NO_x$ conditions (Lambe et al., 2017; Peng et al., 2018). Potential limitations of the method include: (1) Inability to unambiguously deconvolve contributions from multiple oxidants ($O_3$, OH, $NO_3$), which may compete with each other under certain conditions and for specific unsaturated precursors; (2) required use of 254 nm photolysis, which may enhance photolytic losses that compete with OH oxidation, especially for species that are characterized by strong absorption/quantum yield at 254 nm and low OH reactivity (Peng et al., 2016); (3) optimal high-$NO_x$ application at OH exposures corresponding to multiple equivalent days of oxidative aging rather than one day or less.

Here, we adapt a complementary method that uses alkyl nitrite photolysis to generate an alkoxy radical (RO•) and NO. In the presence of air, RO reacts with $O_2$ to generate a carbonyl product (R'O) and a hydroperoxyl ($HO_2$) radical, and NO and $HO_2$ subsequently react to generate OH and $NO_2$. Using this method, $O_3$ is not required to generate OH radicals, and insignificant amounts of $O_3$ or $NO_3$ are generated as byproducts. We present experimental and model characterization of OH and $NO_x$ levels that are generated as a function of photolysis wavelength, and organic nitrite concentration and composition. We furthermore carried out chemical ionization mass spectrometer measurements to compare nitrogen-containing photooxidation products obtained from the reaction of $\alpha$-pinene with radicals generated via alkyl nitrite photolysis or the $O(^1D) + H_2O + N_2O$ reaction.

## 2   Experimental

### 2.1   Alkyl nitrite preparation

Figure 1 shows molecular structures of the alkyl nitrites that were used. Isopropyl nitrite (iPrONO; Pfaltz and Bauer, >95% purity) was used without additional purification. Perdeuterated isopropyl nitrite (iPrONO-$d_7$) and 1,3-propyl dinitrite [1,3-Pr$(ONO)_2$] were synthesized from the action of HONO on isopropanol-$d_8$ or 1,3-propanediol, respectively, as described elsewhere (Noyes, 1933; Andersen et al., 2003; Shuping et al., 2006; Carrasquillo et al., 2014). Briefly, sodium nitrite (>99.999%, Sigma-Aldrich) and the alcohol were combined in a 1.1:1.0 molar ratio and stirred with a magnetic stirrer inside a round bottom flask. Sulfuric acid was added dropwise to the flask – thereby generating HONO upon reaction with sodium nitrite – until a 0.5:1.0 acid:alcohol molar ratio was achieved. The resulting clear yellow liquid was dried over sodium sulfate, neutralized with excess sodium bicarbonate, and then stored in amber vials and refrigerated at $4^oC$ until use (within 1 week of synthesis in this work). Under these storage conditions, the nominal shelf life of iPrONO and similar organic nitrites is approximately 2 years (B. Milburn, personal communication, 29 October 2018).

A syringe pump was used to introduce iPrONO, iPrONO-d$_7$, and 1,3-Pr(ONO)$_2$ through a 10.2 cm length of 0.0152 cm ID teflon tubing at liquid flow rates ranging from 0.016 to 0.63 $\mu$L min$^{-1}$. The liquid organic nitrite was evaporated into a 1 L min$^{-1}$ N$_2$ carrier gas at the end of the tubing. The flow containing organic nitrite vapor was then mixed with a 7 L min$^{-1}$ synthetic air carrier gas at the reactor inlet.

The organic nitrite mixing ratio entering the reactor, $r_{RONO}$, was equal to $\frac{Q_{RONO,g}}{Q_{carrier}}$, where Q$_{RONO,g}$ was the volumetric flow rate of organic nitrite vapor (L min$^{-1}$) and Q$_{carrier}$ was the volumetric flow rate of carrier gas (L min$^{-1}$). Q$_{RONO,g}$ was calculated using the ideal gas law as applied by Liu et al. (2015):

$$Q_{RONO,g} = Q_{RONO,l} \times \frac{\rho}{MW} \times \frac{RT}{P} \times 0.01 \tag{1}$$

where Q$_{RONO,l}$ ($\mu$L min$^{-1}$) is the volumetric flow of organic nitrite liquid, $\rho$ (g cm$^{-3}$) and MW (g mol$^{-1}$) are the organic
nitrite liquid density and molecular weight, $R$ (8.314 J mol$^{-1}$ K$^{-1}$) is the universal gas constant, $T$ (K) is temperature, $P$ (hPa) is pressure, and 0.01 is a lumped pressure, volume and density unit conversion factor.

## 2.2    Alkyl nitrite photolysis

Alkyl nitrites were photolyzed inside a Potential Aerosol Mass (PAM) oxidation flow reactor (Aerodyne Research, Inc.), which is a horizontal 13.3 L aluminum cylindrical chamber (46 cm long $\times$ 22 cm ID) operated in continuous flow mode (Lambe
et al., 2017), with 5.1$\pm$0.3 L min$^{-1}$ flow through the reactor unless stated otherwise. The relative humidity (RH) in the reactor was controlled in the range of 31-63% at 21-32$^\circ$ C using a Nafion humidifier (Perma Pure LLC), with corresponding H$_2$O volumetric mixing ratios of approximately 1.5-1.7%. Four UV lamps centered at $\lambda$ = 254 nm (GPH436T5L; Light Sources, Inc.), 350 nm (F436T5/BL/4P-350; Aerodyne Research, Inc.), or 369 nm (F436T5/BLC/4P-369; Aerodyne Research, Inc.) were used. Emission spectra obtained from the primary manufacturer (Light Sources, Inc. or LCD Lighting, Inc.) are shown in
Figure S1. A fluorescent dimming ballast (IZT-2S28-D, Advance Transformer Co.) was used to regulate current applied to the lamps. The UV irradiance was measured using a photodetector (TOCON-GaP6, sglux GmbH) and was varied by changing the control voltage applied to the ballast between 1.6 and 10 VDC.

NO and NO$_2$ mixing ratios were measured using a NO$_x$ analyzer (Model 405 nm, 2B Technologies), which quantified [NO$_2$] (ppb) from the measured absorbance at $\lambda$ = 405 nm, and [NO] (ppb) by reaction with O$_3$ to convert to NO$_2$. Alkyl nitrites
introduced to the reactor with the lamps turned off consistently generated signals in the both NO and NO$_2$ measurement channels of the NO$_x$ analyzer, possibly due to impurities and/or species generated via iPrONO + O$_3$ reactions inside the analyzer. For example, background NO and NO$_2$ mixing ratios increased from 0 to 1526 ppb and 0 to 1389 ppb as a function of injected [iPrONO] = 0 to 18.7 ppm with the lamps off (Figure S2). We attempted to correct [NO] and [NO$_2$] for this apparent alkyl nitrite interference by subtracting background signals measured in the presence of alkyl nitrite with lamps off, to no avail, because background signals (alkyl nitrite present with lamps off) were large compared to signals obtained with alkyl nitrite present with lamps on. Instead, we constrained [NO] and [NO$_2$] using the photochemical model discussed in Section 2.4.

### 2.2.1 Actinic flux calibration

To quantify the actinic flux $I$ in the reactor for each lamp type, we measured the rate of $NO_2$ photolysis as a function of UV irradiance (Figure S4). Measurements were conducted in the absence of oxygen to avoid $O_3$ formation. The first-order $NO_2$ photolysis rate ($j_{NO_2}$) was calculated using Equation 2:

$$j_{NO_2} = \ln\left(\frac{NO_{2,\tau}}{NO_{2,0}}\right)\frac{1}{\tau_{NO_2}} \tag{2}$$

where $NO_{2,0}$ and $NO_{2,\tau}$ were the steady-state $NO_2$ mixing ratios measured at the exit of the reactor with the lamps turned off and on, respectively. The mean $NO_2$ residence time in the reactor, $\tau_{NO_2}$, was characterized using 10-second pulsed inputs of $NO_2$. To mimic the effect of axial dispersion induced by temperature gradients from the lamps being turned on (Lambe et al., 2011; Huang et al., 2017), residence time distributions were measured in the presence of four lamps centered at $\lambda = 658$ nm (F436T5/4P-658; Aerodyne Research, Inc.), where the $NO_2$ quantum yield is zero (Gardner et al., 1987). $NO_2$ residence time distributions are shown in Figure S3, where $\tau_{NO_2}$ ranged from $120 \pm 34$ s ($\pm 1\sigma$; lamps off) to $98 \pm 63$ s ($\pm 1\sigma$; lamps on) in a manner that is consistent with previous observations (Lambe et al., 2011; Huang et al., 2017). Assuming $\tau_{NO_2} = 98$ s, maximum $j_{NO_2}$ values were 0.12, 0.36, and 0.50 min$^{-1}$ following photolysis at full lamp power at $\lambda = 254$, 350, and 369 nm, respectively.

Corresponding $I_{254}$, $I_{350}$, and $I_{369}$ values were calculated using a photochemical model implemented in the KinSim chemical kinetics solver (Peng et al. (2015); implemented within Igor Pro 7, Wavemetrics Inc.) that incorporated the following reactions:

$$NO_2 + h\nu \rightarrow NO + O(^3P) \tag{R1}$$

$$O(^3P) + NO \rightarrow NO_2 \tag{R2}$$

$$O(^3P) + NO_2 \rightarrow NO_3 \tag{R3}$$

$$O(^3P) + NO_2 \rightarrow NO + O_2 \tag{R4}$$

$NO_2$ absorption cross sections were averaged across the 254, 350, and 369 nm lamp emission spectra, respectively (Table 1) (Atkinson et al., 2004) and input to the model. Maximum $I_{254} = 8.6 \times 10^{16}$ photons cm$^{-2}$ s$^{-1}$, $I_{350} = 6.3 \times 10^{15}$ photons cm$^{-2}$ s$^{-1}$, and $I_{369} = 6.5 \times 10^{15}$ photons cm$^{-2}$ s$^{-1}$, respectively were obtained. While $I_{350}$ and $I_{369}$ values were in agreement with values calculated from lamp manufacturer specifications ($I_{350} = 5.8 \times 10^{15}$ and $I_{369} = 6.2 \times 10^{15}$ photons cm$^{-2}$ s$^{-1}$) within uncertainties, $I_{254}$ obtained from our calibration was ~13 times larger than expected. We hypothesize that this discrepancy was due to the presence of additional minor mercury lines (e.g. $\lambda \sim 313$, 365, 405) that induce $NO_2$ photolysis and that were not fully accounted for using Eq. 2 or the manufacturer spectra (Figure S1). Thus, we instead assume maximum $I_{254} = 6.5 \times 10^{15}$ photons cm$^{-2}$ s$^{-1}$ based on manufacturer specifications.

 ## 2.2.2  OH exposure calibration

The OH exposure ($OH_{exp}$) obtained from alkyl nitrite photolysis, that is, the product of the OH concentration and mean residence time, was calculated from the addition of between 280 and 420 ppb $SO_2$ at the reactor inlet. Over the course of these experiments, $NO_x$ generated from alkyl nitrite photolysis significantly interfered with the $SO_2$ mixing ratio measured with an $SO_2$ analyzer (Model 43i, Thermo Scientific); a representative example is shown in Fig. S5. To circumvent this issue,

we measured the initial $SO_2$ mixing ratio, $[SO_{2,0}]$, prior to alkyl nitrite photolysis, then used an Aerosol Chemical Speciation Monitor (ACSM; Aerodyne Research, Inc.) to measure the concentration of particulate sulfate generated from $SO_2$ + OH reactions.

To relate the measured $[SO_{2,0}]$ and sulfate to $OH_{exp}$, we conducted an offline calibration where 493 ppb $SO_2$ was added to the reactor and OH was generated via $O_3 + h\nu_{254} \rightarrow O(^1D) + O_2$ followed by $O(^1D) + H_2O \rightarrow 2OH$ in the absence of $NO_x$

("OFR254" mode). The reactor was operated at the same residence time and humidity that was used in alkyl nitrite experiments, although we note that humidity will not change the response of the ACSM to sulfuric acid aerosols. Because no particulate ammonia was present aside from trace background levels, we assumed an ACSM collection efficiency of unity for the sulfate particles. $SO_2$ decay and particulate sulfate formation were measured across a range of UV irradiance and $[O_3]$, from which a calibration equation relating sulfate to $OH_{exp}$ was obtained (Figure S6) and applied to alkyl nitrite photolysis experiments.

In a separate experiment conducted with 2.2 ppm of iPrONO input to the reactor at $I_{369} = 6.5 \times 10^{15}$ photons $cm^{-2}$ $s^{-1}$, we verified that the mass of particulate sulfate detected by the ACSM responded linearly to a change in the input mixing ratio of $SO_2$ between 200 and 473 ppb (Figure S7). This suggests that the sulfate particles were large enough for efficient transmission through the inlet lens of the ACSM across the range of $OH_{exp}$ used in our experiments. While not applicable in this work, we note that heterogeneous uptake of $SO_2$ onto organic aerosol may bias OH exposure measurements (Ye et al., 2018).

## 2.3  Chemical Ionization Mass Spectrometer (CIMS) measurements

In a separate set of experiments, mass spectra of gas-phase $\alpha$-pinene photooxidation products were obtained with an Aerodyne high-resolution time-of-flight chemical ionization mass spectrometer using nitrate as the reagent ion ($NO_3^-$-HRToF-CIMS, hereafter abbreviated as $NO_3^-$-CIMS) (Eisele and Tanner, 1993; Ehn et al., 2012). Nitrate ($NO_3^-$) and its higher-order clusters (e.g. $HNO_3NO_3^-$) generated from X-ray ionization of $HNO_3$ were used as the reagent due to their selectivity to highly oxidized

organic compounds, including species that contribute to SOA formation (Ehn et al., 2014; Krechmer et al., 2015; Lambe et al., 2017). The $NO_3^-$-CIMS sampled the reactor output at 10.5 L $min^{-1}$. $\alpha$-Pinene oxidation products were detected as adducts ions of $NO_3^-$. In these experiments, the reactor was operated with a residence time of approximately 80 sec to accommodate the undiluted $NO_3^-$-CIMS inlet flow requirement. OFR369-i(iPrONO) and OFR369-i(iPrONO-d$_7$) were operated using $I_{369} = 6.5 \times 10^{15}$ photons $cm^{-2}$ $s^{-1}$ and >7 ppm alkyl nitrite; in these experiments, $\alpha$-pinene was evaporated into the carrier gas by flowing 1 sccm $N_2$ through a bubbler containing liquid $\alpha$-pinene. Assuming the $N_2$ flow was saturated with $\alpha$-pinene vapor, we estimate ~500 ppb $\alpha$-pinene was introduced to the OFR based on its vapor pressure at room temperature and known dilution ratio into the main carrier gas. In a separate experiment, OFR254-iN$_2$O was operated using $I_{254} = 3.2 \times 10^{15}$ photons $cm^{-2}$

s$^{-1}$ and 5 ppm O$_3$ + 1% H$_2$O + 3.2% N$_2$O. Here, $\alpha$-pinene was introduced by flowing 1 sccm of a gas mixture containing 150 ppm $\alpha$-pinene in N$_2$ into the main carrier gas (this gas mixture was unavailable for the iPrONO photolysis experiments); the calculated $\alpha$-pinene mixing ratio that was introduced to the OFR was $\sim$ 16 ppb.

## 2.4  Photochemical model

We used the KinSim OFR photochemical model to calculate concentrations of radical/oxidant species produced (Peng et al.,
2015; Peng and Jimenez, 2017) . In addition to NO + HO$_2$ $\rightarrow$ OH + NO$_2$ and other reactions included in Peng and Jimenez (2017), the following reactions were added for this study:

$$\mathrm{iPrONO + h\nu \rightarrow iC_3H_7O \bullet + NO} \tag{R5}$$

$$\mathrm{iPrONO + h\nu \rightarrow CH_3CHO + CH_3 \bullet + NO} \tag{R6}$$

$$\mathrm{CH_3 \bullet + O_2 \rightarrow CH_3O_2 \bullet} \tag{R7}$$

$$\mathrm{iC_3H_7O \bullet + O_2 \rightarrow (CH_3)_2C(O) + HO_2} \tag{R8}$$

$$\mathrm{(CH_3)_2CO + OH \rightarrow Products} \tag{R9}$$

$$\mathrm{iPrONO + OH \rightarrow (CH_3)_2C(O) + NO} \tag{R10}$$

$$\mathrm{CH_3CHO + OH \rightarrow CH_3CO \bullet + H_2O} \tag{R11}$$

$$\mathrm{CH_3CO \bullet + O_2 \rightarrow CH_3C(O)O_2 \bullet} \tag{R12}$$

$$\mathrm{CH_3C(O)O_2 \bullet + NO \rightarrow CH_3 \bullet + NO_2 + CO_2} \tag{R13}$$

$$\mathrm{CH_3O_2 \bullet + NO \rightarrow CH_3O \bullet + NO_2} \tag{R14}$$

$$\mathrm{CH_3O \bullet + O_2 \rightarrow HCHO + HO_2} \tag{R15}$$

$$\mathrm{HCHO + OH \rightarrow HCO \bullet + H_2O} \tag{R16}$$

$$\mathrm{HCO \bullet + O_2 \rightarrow CO + HO_2} \tag{R17}$$

Model input parameters included pressure, temperature, [H$_2$O], [iPrONO], mean residence time, actinic flux, and absorption cross sections and bimolecular rate constants shown in Table 1. We assumed the quantum yield of Reaction R5 to be 0.50 above 350 nm (Raff and Finlayson-Pitts, 2010). We assumed the quantum yield of Reaction R6 to be 0.04 above 350 nm (value for $t$-butyl nitrite) (Calvert and Pitts, 1966), suggesting minimal influence of CH$_3$O$_2$ and CH$_3$C(O)O$_2$ that are generated via Reactions R7, R10, and R11 following iPrONO decomposition to CH$_3$ and CH$_3$CHO via Reaction R6. At 254 nm, the
influence of CH$_3$O$_2$ and CH$_3$C(O)O$_2$ on ensuing photochemistry may be more significant.This is due to a higher quantum yield of Reaction R6 at 254 nm, which is estimated to be 0.86 under vacuum (Calvert and Pitts, 1966). Assuming that all 254 nm photons initiate photolysis, the quantum yield of Reaction R5 is 0.14. Due to collisional deactivation at 1 atm that prevents $i$-C$_3$H$_7$O$\bullet$ decomposition, the quantum yield of Reaction R5 at $\lambda$ = 254 nm and 1 atm is expected to be higher than 0.14. Because quantum yield measurements were unavailable at these conditions, we applied an upper limit quantum yield of 0.50

as applicable at $\lambda > 350$ nm and 1 atm (Raff and Finlayson-Pitts, 2010). We calculated a corresponding nominal quantum yield of 0.32 by averaging the lower and upper limit values of 0.14 and 0.50, resulting in a quantum yield of 0.68 for Reaction R6.

We assumed that the residence time distribution of iPrONO in the reactor was similar to the residence time distribution of $NO_2$. To model iPrONO photolysis at $\lambda = 254$ nm, we extended the range of previously-measured $\sigma_{iPrONO}$ values by measuring the gas phase absorption cross sections of iPrONO (purified via four freeze-pump-thaw cycles prior to measurement) down to $\lambda = 220$ nm using a custom-built absorption cell (Raff and Finlayson-Pitts, 2010). Results at $\lambda = 220$ to 436 nm are shown in Figure S1 and are in agreement with previous work (Raff and Finlayson-Pitts, 2010) over the range of overlap at $\lambda = 300$ to 450 nm.

To account for uncertainties associated with the assumptions we made for quantum yield values, as well as uncertainties in other kinetic parameters, temperature, residence time, actinic flux, and organic nitrite concentration, we performed Monte Carlo uncertainty propagation (BIPM et al., 2008) as described previously (Peng et al., 2015; Peng and Jimenez, 2017). All uncertain kinetic parameters were assumed to follow log-normal distributions unless stated otherwise below. Uncertainties in rate constants and cross sections newly included in this study were adopted from Burkholder et al. (2015) if available. The relative uncertainty in the rate constant of Reaction R13 was estimated to be 40% based on the dispersion of rate constant measurements of published $RO_2 + NO$ reactions. We assumed the random samples of the quantum yields of Reactions R5 and R6 at 254 nm and Reaction R6 at 369 nm followed uniform distributions in the range of [0.50, 0.86], [0.14, 0.50] and [0, 0.20], respectively. We assumed uncertainties of 5 K and 20 s in temperature and residence time (normal distributions assumed), respectively, and relative uncertainties of 50%, 100%, and 25% in actinic flux at 369 nm, actinic flux at 254 nm, and organic nitrite concentration, respectively.

## 3  Results and Discussion

We first characterized $OH_{exp}$ and $NO_x$ by separately varying the photolysis wavelength (Sect. 3.1.1) and input organic nitrite concentration to the reactor (Sec. 3.1.2), with the goal of identifying optimal OFR conditions for OH and $NO_x$ generation via iPrONO photolysis. Second, we synthesized novel alkyl nitrites and compared their performance to iPrONO (Sec. 3.2). Third, we parameterized $OH_{exp}$ and $NO_2$ production in a set of algebraic equations to guide selection of OFR experimental conditions. Finally, we compared $NO_3^-$-CIMS spectra of photooxidation products generated from reaction of $\alpha$-pinene with radicals produced via alkyl nitrite photolysis and $O(^1D) + H_2O + N_2O$ reactions.

### 3.1  $OH_{exp}$ and $NO_x$ generated from iPrONO photolysis

#### 3.1.1  Effect of photolysis wavelength

Figure 2 shows $OH_{exp}$, [NO], and [$NO_2$] obtained as a function of actinic flux following photolysis of 1.9 ppm of iPrONO injected into the reactor at $\lambda = 254$, 350, or 369 nm. These systems are hereafter designated as OFR254-i(iPrONO), OFR350-i(iPrONO), and OFR369-i(iPrONO), respectively; similar nomenclature is adapted for other alkyl nitrites. In these notations,

the numbers following "OFR" are the photolysis wavelengths (in nm), and the "i" preceding the parentheses means initial injection of the radical precursor compound noted in the parentheses. Modeled $OH_{exp}$, NO, and $NO_2$ values for the OFR254-i(PrONO) and OFR369-i(PrONO) modes are shown in Figure 2 at the same nominal operating conditions.

At a fixed photolysis wavelength, $OH_{exp}$, NO, and $NO_2$ increased with increasing actinic flux. Measured and modeled $OH_{exp}$ values were in agreement within uncertainties at $\lambda = 369$ nm. At $\lambda = 254$ nm, model $OH_{exp}$ results were higher than the measurements, perhaps due to uncertainty in assumptions that were necessary to model OFR254-i(iPrONO) (Section 2.4). Higher $NO_2$ concentrations were modeled at $\lambda = 254$ nm than at $\lambda = 369$ nm because more iPrONO was photolyzed and the $NO_2$ yield was only weakly dependent on the fate of $i\text{-}C_3H_7O\bullet$. For example, NO is converted to $NO_2$ either via reaction with $HO_2$ obtained via Reaction R5 or $CH_3O_2\bullet$ and $CH_3C(O)O_2\bullet$ obtained via Reaction R6. However, the effect of photolysis wavelength on NO and $OH_{exp}$ was different. Specifically, the highest NO concentration and $OH_{exp}$ was achieved via OFR369-i(iPrONO). $OH_{exp}$ achieved via OFR369-i(iPrONO) was slightly higher than $OH_{exp}$ attained using OFR350-i(iPrONO), likely because photolysis of both iPrONO and $NO_2$, whose reaction with OH suppresses $OH_{exp}$, is more efficient at $\lambda = 369$ nm than at $\lambda = 350$ nm (Figure S1 and Table 1). Further, the NO and OH yields achieved via OFR254-i(iPrONO) were suppressed due to significant (>73%) decomposition of $iC_3H_7O\bullet$ (Calvert and Pitts, 1966). The dependence of OH, NO and $NO_2$ on the quantum yields of Reactions R5 and R6 was confirmed by sensitivity analysis of uncertainty propagation inputs and outputs as described in Section 2.4. $OH_{exp}$ and NO were strongly anticorrelated with the quantum yield of Reaction R6, whereas the correlation between $NO_2$ and the quantum yield of Reaction R6 was negligible.

The products of this decomposition, i.e., $CH_3CHO$ and $CH_3\bullet$, both have adverse effects with regard to our experimental goals: $CH_3CHO$ is reactive toward OH and can thus suppress OH; the $RO_2\bullet$ formed through this reaction, $CH_3C(O)O_2\bullet$, consumes NO and generates $NO_2$ but does not generate OH; $CH_3\bullet$ rapidly converts to $CH_3O_2\bullet$, which also consumes NO and generates $NO_2$ but does not directly produce OH. Importantly, Figure 2 suggests that it is preferable to photolyze alkyl nitrates at $\lambda > 350$ because optimal $OH_{exp}$ and $NO:NO_2$ were attained via OFR369-i(iPrONO). Moreover there is added risk for significant unwanted photolysis of organics via OFR254-i(iPrONO) (Peng et al., 2016).

### 3.1.2 Effect of alkyl nitrite concentration

Figure 3 shows measured $OH_{exp}$ and modeled $NO_x$ concentrations obtained from photolysis of 0.5 to 20 ppm iPrONO at $I_{369}$ $\approx 7 \times 10^{15}$ photons $cm^{-2}$ $s^{-1}$. [NO] and [$NO_2$] increased with increasing [iPrONO], as expected. For [iPrONO] $\leq 5$ ppm, $OH_{exp}$ increased with increasing [iPrONO] because the rate of OH production increased faster than the rate of OH destruction from reaction with iPrONO and $NO_2$. The model results showed that for [iPrONO] $> 5$ ppm, the opposite was true and $OH_{exp}$ plateued or decreased. A maximum $OH_{exp} = 7.8 \times 10^{10}$ molecules $cm^{-3}$ s was achieved via photolysis of 10 ppm iPrONO, with corresponding modeled [NO] and [$NO_2$] values of 148 and 405 ppb respectively. Modeled $NO_3$ concentrations were negligible in OFR369-i(iPrONO) ($\leq 1$ ppt) because there was no $O_3$ present and $NO_3$ production via $HNO_3 + OH \rightarrow NO_3 + H_2O$ reactions was insignificant.

## 3.2 $OH_{exp}$ generated from photolysis of perdeuterated iPrONO and 1,3-propyl dinitrite

Although $OH_{exp} = 7.8 \times 10^{10}$ molecules $cm^{-3}$ s (approximately 0.6 d of equivalent atmospheric OH exposure) may be suitable for some OFR applications, it may be insufficient to simulate multigenerational oxidative aging of precursors with OH rate constants slower than $\sim 10^{-11}$ $cm^3$ molecule$^{-1}$ s$^{-1}$. We attempted to synthesize three $C_3$ alkyl nitrites that we hypothesized could generate higher $OH_{exp}$ than iPrONO: perdeuterated isopropyl nitrite (iPrONO-d$_7$), 1,3-propyl dinitrite [1,3-Pr(ONO)$_2$], and hexafluoroisopropyl nitrite (HFiPrONO). We successfully synthesized 1,3-Pr(ONO)$_2$ and iPrONO-d$_7$, but were unable to synthesize HFiPrONO (Sect. 3.5.3). Figure 4 shows $OH_{exp}$ attained from photolysis of 1.2 ppm 1,3-Pr(ONO)$_2$ and 1.7 ppm iPrONO-d$_7$ as a function of $I_{369}$, along with the model output for OFR369-i(iPrONO) shown for reference. At these organic nitrite concentrations and $I_{369}$ values, maximum $OH_{exp}$ measurements were: $1.1 \times 10^{11}$ (iPrONO-d$_7$), $4.0 \times 10^{10}$ (iPrONO), and $1.8 \times 10^{10}$ molecules $cm^{-3}$ s [1,3-Pr(ONO)$_2$], respectively. At maximum $I_{369}$ and after correcting for the different iPrONO, iPrONO-d$_7$ and 1,3-Pr(ONO)$_2$ concentrations that were used (Figure 3), $OH_{exp,iPrONO-d_7} \approx 2.9 \times OH_{exp,iPrONO}$ and $OH_{exp,1,3-Pr(ONO)_2} \approx 0.81 \times OH_{exp,iPrONO}$.

We hypothesize that higher $OH_{exp}$ obtained from OFR369-i(iPrONO-d$_7$) relative to OFR369-i(iPrONO) was due to $\sim 2.6$ times lower OH reactivity of iPrONO-d$_7$ relative to iPrONO (Nielsen et al., 1988, 1991) and 6 times lower OH reactivity of acetone-d$_6$ relative to acetone (Raff et al., 2005). This hypothesis is supported by the modeled $OH_{exp}$ attained via OFR369-i(iPrONO-d$_7$), which is in agreement with measured $OH_{exp}$ within uncertainties and is 41% higher than modeled $OH_{exp}$ attained via OFR369-i(iPrONO). Model simulations revealed that this effect was most pronounced near the reactor inlet (e.g. at low residence time), where the local OH concentration was higher than elsewhere in the reactor because $NO_x$ was very low, resulting in higher sensitivity of [OH] to the OH reactivity of the specific organic nitrite that was used. On the other hand, OFR369-i(1,3-Pr(ONO)$_2$) was less efficient than OFR369-i(iPrONO). In this case, it is possible that higher $NO_2$ production during 1,3-Pr(ONO)$_2$ photolysis and/or production of more reactive intermediates (e.g. malonaldehyde) offset any benefit gained from faster OH production via photolysis of both -ONO groups or more efficient photolysis of one -ONO group (Wang and Zu, 2016).

## 3.3 $OH_{exp}$ and $NO_2$ estimation equations for OFR369-i(iPrONO) and OFR369-i(iPrONO-d$_7$)

Previous studies reported empirical $OH_{exp}$ algebraic estimation equations for OFR185 and OFR254 (Li et al., 2015; Peng et al., 2015).These equations parameterize $OH_{exp}$ as a function of readily-measured experimental parameters, therefore providing a simpler alternative to detailed photohemical models that aids in experimental planning and analysis. Here, we expand on those studies by deriving $OH_{exp}$ and $NO_2$ estimation equations for OFR369-i(iPrONO) and OFR369-i(iPrONO-d$_7$). Model results (14641 model runs in total) obtained from the base case of the model ($SO_2$ as surrogate of external OH reactivity, "$OHR_{ext}$") were used to derive the following equations that allow estimating the OH exposure for OFR369-i(iPrONO) and OFR369-i(PrONO-d$_7$):

$$\log(\mathrm{OH_{exp}}) = \log(\mathrm{I_{369}}) - 0.0026728 \star \mathrm{OHR_{ext}} + 0.46017 \star \log([\mathrm{iPrONO}]) + 1.1928 \star \log(\tau) \tag{3}$$

$$+ 0.35317 \star \log([\mathrm{iPrONO}]) \star \log(\mathrm{OHR_{ext}}) - 0.11109 \star \log(\mathrm{OHR_{ext}}) \star \log(\tau) \tag{4}$$

$$- 0.015606 \star \log(\mathrm{I_{369}}) \star \log([\mathrm{iPrONO}]) \star \log(\tau) - 7.6164 \tag{5}$$

$$\log(\mathrm{OH_{exp}}) = 0.85558 \star \log(\mathrm{I_{369}}) - 0.0029546 \star \mathrm{OHR_{ext}} + 0.61837 \star \log([\mathrm{iPrONO\text{-}d_7}]) \tag{6}$$

$$+ 1.2115 \star \log(\tau) + 0.36081 \star \log([\mathrm{iPrONO\text{-}d_7}]) \star \log(\mathrm{OHR_{ext}}) \tag{7}$$

$$- 0.15501 \star \log(\mathrm{OHR_{ext}}) \star \log(\tau) - 0.017061 \star \log(\mathrm{I_{369}}) \star \log([\mathrm{iPrONO\text{-}d_7}]) \star \log(\tau) \tag{8}$$

$$- 5.1541 \tag{9}$$

where $\mathrm{OH_{exp}}$, $I_{369}$, $\mathrm{OHR_{ext}}$, [iPrONO or iPrONO-$d_7$], and $\tau$ are in units of molecules cm$^{-3}$ s, photons cm$^{-2}$ s$^{-1}$, s$^{-1}$, ppm, and s, respectively. Fit coefficients were obtained by fitting Equations 3 and 6 to $\mathrm{OH_{exp}}$ model results over the following range of OFR parameters: ([iPrONO/iPrONO-$d_7$]; 0.2-20 ppm), $I_{369}$ ($1 \times 10^{15}$ - $2 \times 10^{16}$ photons cm$^{-2}$ sec$^{-1}$), $\mathrm{OHR_{ext}}$ (1-200 s$^{-1}$), and residence time, $\tau$, between 30 and 200 sec. We explored 11 logarithmically evenly distributed values in these ranges for each parameter, and thus performed simulations for 14641 model cases in total. To determine the functional form of Equations 3 and 6, we used the sum of the logarithms of first-, second- and third-order terms of the four parameters and iteratively removed the terms with very small fit coefficients until further removal of remaining terms significantly worsened the fit quality.

Figures 5a and 5c compare $\mathrm{OH_{exp}}$ estimated from Equations 3 and 6 and calculated from the model described in Sect. 2.4. The mean absolute value of the relative deviation is 29%, indicating that the estimation equations are typically producing results within the inherent model uncertainties. Care should be taken to not use the equations away from the range of which they were derived, as much larger errors are possible when extrapolating.

While several techniques are available to monitor $NO_2$, interferences from other nitrogen-containing species are well known and may create issues similar to those shown in Figure 2f. $NO_2$ production and loss rates are primarily governed by the alkyl nitrite concentration, actinic flux, and residence time in the OFR. These parameters were experimentally constrained (Section 2.2.2). Thus, we derived $NO_2$ estimation equations for OFR369-i(iPrONO) (Eq. 10) and OFR369-i(iPrONO-$d_7$) (Eq. 11) as a function of [RONO], $I_{369}$, and $\tau$, to all of which $NO_2$ production is proportional, over the same phase space used to fit Equations 3 and 6:

$$\log(\mathrm{NO_2}) = \log(\mathrm{I_{369}}) + \log([\mathrm{iPrONO}]) + \log(\tau) - 6.2198 \tag{10}$$

$$\log(\mathrm{NO_2}) = \log(\mathrm{I_{369}}) + \log([\mathrm{iPrONO\text{-}d_7}]) + \log(\tau) - 6.2607 \tag{11}$$

Figures 5b and 5d compare $NO_2$ estimated from Equations 3 and 6 and calculated from the model described in Sect. 2.4. The mean absolute value of the relative deviation between $NO_2$ estimated by Equations 10 and 11 and $NO_2$ computed by the photochemical model is 19%. The mean model $NO:NO_2$ fraction is approximately 0.33 (Figures 2-3).

## 3.4 $NO_3^-$-CIMS spectra of organic nitrates generated from $\alpha$-pinene + OH/OD reactions via OFR369-i(iPrONO), OFR369-i(iPrONO-$d_7$), and OFR254-iN$_2$O

To evaluate the efficacy of OFR369-i(iPrONO), OFR369-i(iPrONO-$d_7$), and OFR254-iN$_2$O for generating $HO_x$ under high-$NO_x$ photooxidation conditions, we obtained $NO_3^-$-CIMS spectra of $\alpha$-pinene + OH/OD nitrogen-containing oxidation products generated using each method, with experimental conditions described in Sect. 2.3. Calculated OH exposures for OFR369-i(iPrONO), OFR369-i(iPrONO-$d_7$), and OFR254-iN$_2$O were $2.9\times10^{10}$, $5.9\times10^{10}$ and $5.0\times10^{11}$ molecules cm$^{-3}$ s, respectively, in the absence of OH consumption due to $\alpha$-pinene. These calculated steady-state OH$_{exp}$ values decreased to $8.5\times10^8$, $6.8\times10^8$ and $4.6\times10^{11}$ molecules cm$^{-3}$ s after accounting for OH consumption. This suggests that most of the OH that was produced in these OFR369-i(iPrONO/iPrONO-$d_7$) experiments was consumed by $\alpha$-pinene and its early-generation photooxidation products. OH suppression relative to 254 nm photons, $O_3$, and O is not a concern in OFR369-i(iPrONO), unlike in OFR254-iN$_2$O (Peng et al., 2016).

Exposure of $\alpha$-pinene to OH/OD generated via OFR369-iPrONO, OFR369-iPrONO-$d_7$, and OFR254-iN$_2$O produced $C_7$ - $C_{10}$ organic nitrate and $C_{10}$ dinitrate signals that are shown in Figures 6a-d. ( $[(NO_3)C_7H_9NO_8]^-$ and $[(NO_3)C_7H_{11}NO_8]^-$ signals at $m/z$ = 297 and 299 are excluded due to significant intra- and inter-experiment variability for unknown reasons). Figure 6 shows that many of the same products are observed independent of radical precursor. The Figure 6a spectrum (OFR369-i(iPrONO)) is shifted to lower oxygen-to-carbon ratio relative to Figures 6b (OFR254-iN$_2$O) and Figure 6c (OFR369-i(iPrONO-$d_7$)) , consistent with the lower OH$_{exp}$ achieved with OFR369-i(iPrONO) compared to OFR369-i(iPrONO-$d_7$) and OFR254-iN$_2$O. For example, $[(NO_3)C_{10}H_{15}NO_7]^-$ was the largest $C_{10}$ nitrate signal observed via OFR369-i(iPrONO), whereas $[(NO_3)C_{10}H_{15}NO_8]^-$ was the largest $C_{10}$ nitrate signal observed via OFR369-i(iPrONO-$d_7$) and OFR254-iN$_2$O. Qualitatively similar trends were observed for $C_7$ - $C_9$ organic nitrates and $C_{10}$ dinitrates across the three systems.

Two additional features are of note in Figure 6. First, a series of ion signals at $m/z$ = 312, 328, 344, 360, 376, 392, 408 and 340, 356, 372, 388, 402, 420 were observed at higher levels via OFR369-i(iPrONO-$d_7$) relative to OFR369-i(iPrONO). These ions are plotted separately in Figure 6d. The most plausible explanation is the additional contribution of $[(NO_3)C_8H_{10}DNO_{8-14}]^-$ and $[(NO_3)C_{10}H_{14}DNO_{7-14}]^-$ ions that retain -OD functionality following initial addition of OD (rather than OH) to $\alpha$-pinene. There is evidence of other deuterium-containing ions in Figure 6b that are either less prominent or more difficult to resolve from other ions at the same integer mass. Second, $C_{10}$ dinitrates were present in all three spectra, with the highest dinitrate fractions observed in Figures 6b (0.090) and 6c (0.081), and the lowest dinitrate fraction observed in Figure 6a (0.056). Dinitrates are presumably generated from $\alpha$-pinene following (1) two OH reactions followed by two $RO_2$ + NO termination reactions or (2) one $NO_3$ reaction followed by one $RO_2$ + NO termination reaction. Previous application of OFR254-iN$_2$O could not exclude the contribution of $\alpha$-pinene + $NO_3$ reactions, with $NO_3$ radicals generated from $NO_2$ + $O_3$ and other

reactions (Lambe et al., 2017). However, generation of dinitrates via OFR369-i(iPrONO-d$_7$), which produced negligible NO$_3$, suggests that dinitrates are not an artifact of unwanted $\alpha$-pinene + NO$_3$ reactions.

The ability of OFR369-i(iPrONO) and OFR369-i(iPrONO-d$_7$) to mimic polluted atmospheric conditions can be evaluated by comparing signals observed in Figures 6a-c with NO$_3^-$-CIMS spectra obtained in Centreville, AL, USA (Massoli et al., 2018) and in Hyytiala, Finland (Yan et al., 2016) that are shown in Figures 6e-f. Both measurement locations are influenced by local biogenic emissions mixed with occasional anthropogenic outflow. Figures 6e and 6f were obtained on 25 June 2013 (7:30–11:00 Centreville time) and 11 April 2012 (10:00-13:00 Hyytiala time) respectively. The mean NO mixing ratios during these periods were 0.53±0.17 (Centreville) and 0.27±0.09 ppb (Hyytiala). In Centreville, a "terpene nitrate" source factor that peaked during the early morning contained C$_9$H$_{15}$NO$_{5-7}$, C$_{10}$H$_{15,17}$NO$_{6-10}$, and C$_{10}$H$_{14,16}$N$_2$O$_{8-11}$ compounds (Massoli et al., 2018). The largest C$_{10}$ nitrate and dinitrate species in that factor were C$_{10}$H$_{15}$NO$_6$, C$_{10}$H$_{15}$NO$_8$, C$_{10}$H$_{16}$N$_2$O$_9$ and C$_{10}$H$_{16}$N$_2$O$_{10}$. In Hyytiala, concentrations of C$_{10}$H$_{15}$NO$_{7-11}$ and C$_{10}$H$_{16}$N$_2$O$_{8-11}$ peaked in the morning and early afternoon. Elevated C$_{10}$ dinitrate levels during the daytime in Hyytiaila (Figure 6f suggest their formation from monoterpenes following two OH reactions followed by two RO$_2$ + NO termination reactions, as proposed earlier.

Overall, Figure 6 shows that many of the C7-C10 nitrogen-containing compounds observed in Centreville and Hyytiala were generated via OFR369-i(iPrONO), OFR369-i(iPrONO-d$_7$) and OFR254-iN$_2$O. Due to the local nature of the ambient terpene emissions at the Centreville and Hyytiala sites, the associated photochemical age was presumably <1 day. Thus, while the ambient NO$_3^-$-CIMS spectra at those sites were more complex and contained contributions from precursors other than $\alpha$-pinene, the oxidation state of the ambient terpene-derived organic nitrates was more closely simulated via OFR369-i(iPrONO) or OFR369-i(iPrONO-d$_7$), where the largest C$_{10}$ nitrates and dinitrates were C$_{10}$H$_{15}$NO$_7$ and C$_{10}$H$_{16}$N$_2$O$_9$ (OFR369-i(iPrONO); Figure 6a), and C$_{10}$H$_{15}$NO$_8$, C$_{10}$H$_{15}$NO$_9$ and C$_{10}$H$_{16}$N$_2$O$_9$ (OFR369-i(iPrONO-d$_7$); Figure 6c). By comparison, C$_{10}$H$_{15}$NO$_8$ and C$_{10}$H$_{16}$N$_2$O$_{11}$ were the largest nitrate and dinitrate species generated via OFR254-iN$_2$O (Figure 6b).

## 3.5 Anticipated performance of alternative high-NO$_x$ HO$_x$ precursors in OFRs

### 3.5.1 Methyl nitrite (MeONO)

MeONO is commonly used as an OH radical source in environmental chamber studies (Atkinson et al., 1981; Matsunaga and Ziemann, 2010; Chhabra et al., 2011; Finewax et al., 2018). To evaluate its potential use in OFRs, we examined previous measurements in an environmental chamber equipped with blacklights (j$_{NO_2}$ = 0.27 min$^{-1}$, assumed 350 nm wavelength), where photolysis of 10 ppm MeONO generated [OH] $\sim$2$\times$10$^8$ molecules cm$^{-3}$ for a few minutes (Atkinson et al., 1981). In our OFR, $j_{NO_2,max}$ = 0.36 min$^{-1}$ at $\lambda$ = 350 nm. Thus, over 98 sec exposure time, we anticipate OH$_{exp}$ $\approx$ 2$\times$10$^{10}$ molecules cm$^{-3}$ s would be obtained via photolysis of 10 ppm MeONO in OFRs. This is lower than the OH$_{exp}$ attained via photolysis of 10 ppm iPrONO even after correcting for different $j_{NO_2}$ values in the different studies. Lower OH$_{exp}$ achieved from MeONO photolysis is presumably due to the higher reactivity of formaldehyde, the primary photolysis product of MeONO, relative to acetone, the primary photolysis product of iPrONO at 369 nm (Raff and Finlayson-Pitts, 2010). Along with less efficient OH

production, MeONO must be synthesized, trapped at low temperature, and stored under vacuum. Thus, there is no advantage to using OFR350-iMeONO (or OFR350-MeONO-d$_4$) in OFRs relative to OFR369-i(iPrONO) or OFR369-i(iPrONO-d$_7$).

### 3.5.2 Nitrous acid (HONO)

HONO is also commonly used as an OH radical source in environmental chamber studies. To evaluate its potential application in OFRs, we examined previous measurements in an environmental chamber equipped with blacklights, where photolysis of

3-20 ppm HONO generated initial [OH] $\approx 6 \times 10^7$ molecules cm$^{-3}$ (Cox et al., 1980), which is 3.3 times lower than [OH] obtained from photolysis of comparable levels of MeONO (Section 3.5.1). Lower OH$_{exp}$ achieved from HONO photolysis is presumably due to higher OH reactivity of HONO relative to MeONO/iPrONO. Additionally, HONO is difficult to prepare without NO$_2$ impurities (Febo et al., 1995) that may cause additional OH suppression. For these reasons, we believe that there is no advantage to using HONO as a HO$_x$ precursor in OFRs.

### 3.5.3 Hexafluoroisopropyl nitrite (HFiPrONO)

HFiPrONO has been synthesized from O-nitrosation of hexafluoroisopropanol (Andersen et al., 2003; Shuping et al., 2006). We predict that OFR369-i(HFiPrONO) should attain higher OH$_{exp}$ than OFR369-i(iPrONO) and OFR369-i(iPrONO-d$_7$) due to similar photolysis rates (Andersen et al., 2003) and ∼200 times lower OH reactivity of HFiPrONO/hexafluoroacetone relative to iPrONO/acetone (Atkinson et al., 1992; Tokuhashi et al., 1999). Simple modeling calculations suggest that application

of OFR369-i(HFiPrONO) may achieve up to a week of equivalent OH exposure. We made several unsuccessful attempts to synthesize HFiPrONO and other fluorinated alkyl nitrites with a procedure similar to that used by Andersen et al. (2003) and Shuping et al. (2006). The synthesis product was blue (not yellow) in color when trapped or stored in nitrogen, generated negligible OH upon irradiation in the reactor, and evolved into brown vapor in the presence of air or upon warming to room temperature (Figure S8). These observations suggest the formation of N$_2$O$_3$, which we hypothesize was formed in solution from the reactions

$$H_2SO_4 + 2NaNO_2 \rightarrow 2HONO + Na_2(SO_4) \tag{R18}$$

$$2HONO \rightarrow NO + NO_2 + H_2O \tag{R19}$$

$$NO + NO_2 \rightleftharpoons N_2O_3 \tag{R20}$$

This pathway may have been favored if the O-nitrosation of hexafluoroisopropanol was slow compared to non-fluorinated alcohols.

### 4 Conclusions

Recently, we developed new methods that enable NO$_x$-dependent photooxidation studies in OFRs using O($^1$D) + N$_2$O + H$_2$O reactions via O$_3$ photolysis at $\lambda$ = 254 nm and/or H$_2$O + N$_2$O photolysis at 185 nm (OFR254-iN$_2$O and OFR185-

iN$_2$O) (Lambe et al., 2017; Peng et al., 2018). Alkyl nitrite photolysis is an established method that facilitates high-NO$_x$ photooxidation studies in modern OFRs. Here, we adapted alkyl nitrite photolysis for new OFR applications by characterizing the photolysis wavelength, nitrite concentration, and nitrite composition that result in optimal HO$_x$ and NO$_x$ generation capabilities. Based on our results, we recommend photolysis of 5-10 ppm isopropyl nitrite at $\lambda \approx 365$ - 370 nm photolysis wavelength and $I > 10^{15}$ photons cm$^{-2}$ s$^{-1}$. If the user has the resources to synthesize iPrONO-d$_7$, better performance is

expected relative to iPrONO. Alkyl nitrite photolysis at $\lambda = 254$ nm is not recommended. Taken together, OFR254/185-iN$_2$O and OFR369-i(iPrONO/iPrONO-d$_7$) are complementary methods that provide additional flexibility for NO$_x$-dependent OFR studies. OFR254/OFR185-iN$_2$O and OFR185-iN$_2$O generate variable-NO$_x$ photooxidation conditions (NO:HO$_2 \approx 0$ - 100) and are suitable for to the characterization of multigenerational oxidative aging processes at up to OH$_{exp} \approx (5\text{-}10)\times10^{11}$ molecules cm$^{-3}$ s ($\sim$ 5-10 equivalent days). OFR369-i(iPrONO)/OFR369-i(iPrONO-d$_7$) generate high-NO photooxidation

conditions (NO:HO$_2 \approx 10$ - 10000; NO:NO$_2 \approx 0.2\text{-}0.7$) with minimal O$_3$ and NO$_3$ formation at longer photolysis wavelength than OFR254/185-iN$_2$O. We anticipate that alkyl nitrite photolysis is advantageous for the characterization of first-generation, high-NO$_x$ photooxidation products of most precursors at up to OH$_{exp} \approx 1\times10^{11}$ molecules cm$^{-3}$ s (1 equivalent day), which is comparable to environmental chambers investigating high-NO$_x$ conditions. The generation of OD (rather than OH) via OFR369-i(iPrONO-d$_7$) may be useful in photooxidation studies of unsaturated precursors due to the shift on the *m/z* of the addition products, though at the potential expense of generating more complex distributions of oxidation products. Potential disadvantages of the OFR369-i(iPrONO) method are: (1) restriction to high-NO photochemical conditions; (2) restriction to OH$_{exp}$ of 1 equivalent day or less; (3) additional complexity involved with integration of the alkyl nitrite source (compared to O$_3$ + H$_2$O + N$_2$O); (4) additional cost and complexity to retrofit a specific OFR design with blacklights; (5) it acts as an

interference that precludes NOx measurements by chemiluminescence detection. Future work will evaluate the ability of each method to mimic polluted atmospheric conditions in specific source regions.

*Data availability.*  Data presented in this manuscript is available upon request.

*Competing interests.*  The authors declare no competing interests.

*Acknowledgements.*  This research was supported by the Atmospheric Chemistry Program of the US National Science Foundation under

grants AGS-1536939, AGS-1537446, AGS-1537009, and AGS-1352375 (to JDR). ZP and JLJ were supported by DOE (BER/ASR) DE-SC0016559 and NOAA NA18OAR4310113. ATL thanks Paola Massoli, Penglin Ye, and Phil Croteau (ARI) for experimental assistance, and Chao Yan (University of Helsinki), William Brune (Penn State), Pengfei Liu (Harvard), Wai Yip Fan (National University of Singapore), Manjula Canagaratna (ARI), John Jayne (ARI), Charles Kolb (ARI), and Paul Ziemann (CU Boulder) for helpful discussions.

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

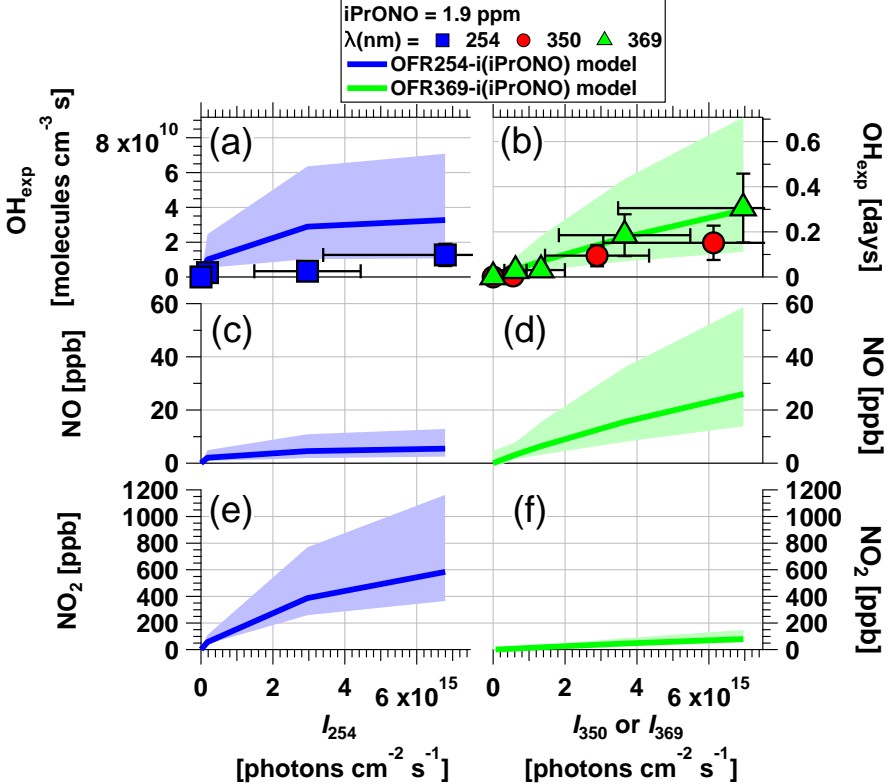

**Figure 1.** Molecular structures of isopropyl nitrite, isopropyl nitrite-d₇ (d = deuterium = ²H), and 1,3-propyl dinitrite.

**Figure 2.** Measured and modeled (a-b) OH exposure, (c-d) NO mixing ratio, and (e-f) $NO_2$ mixing ratio values as a function of actinic flux (*I*) following photolysis of 1.9 ppm isopropyl nitrite (iPrONO) at $\lambda$ = 254 (OFR254-i(iPrONO)), 350, or 369 nm (OFR369-i(iPrONO)) in the PAM oxidation flow reactor. Error bars for measurements represent $\pm50\%$ uncertainty in $OH_{exp}$ and *I*-values.

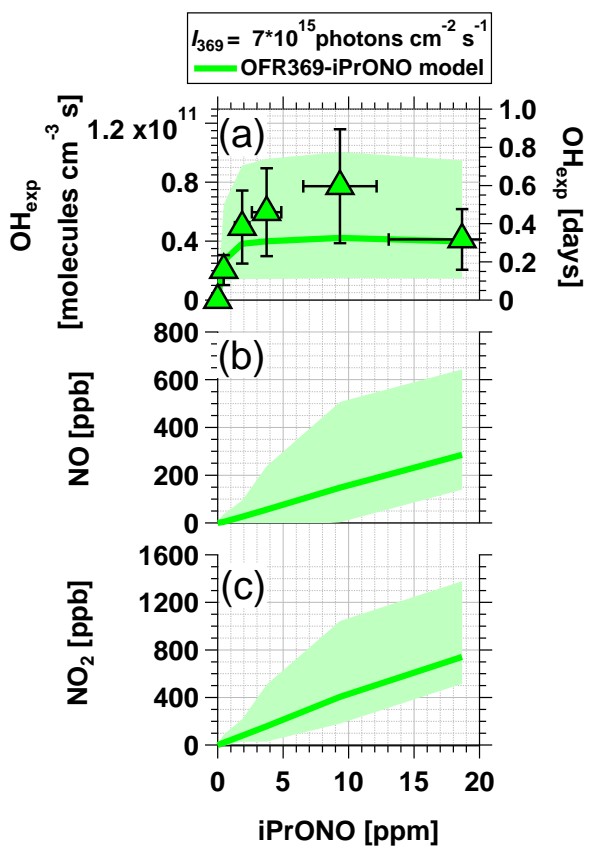

**Figure 3.** Measured and modeled (a) OH exposure, (b) NO mixing ratio and (c) $NO_2$ mixing ratio values obtained using OFR369-i(PrONO) at $I_{369} = 7 \times 10^{15}$ ph cm$^{-2}$ sec$^{-1}$ as a function of iPrONO mixing ratio. Error bars for measurements represent $\pm 50\%$ uncertainty in $OH_{exp}$ and estimated $\pm 30\%$ uncertainty in iPrONO mixing ratio values.

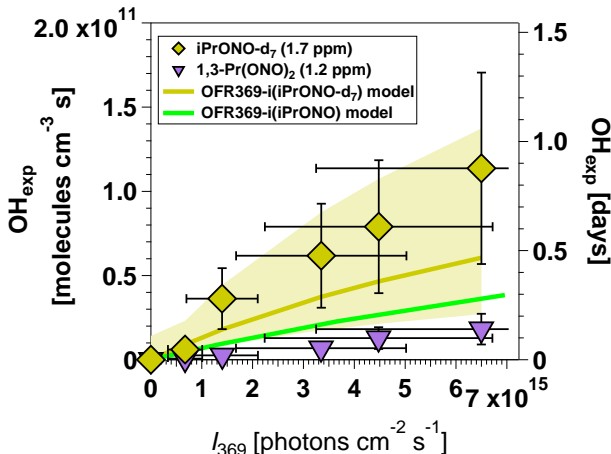

**Figure 4.** Measured and modeled OH exposure values measured as a function of $I_{369}$ following photolysis of perdeuterated isopropyl nitrite (iPrONO-$d_7$) and 1,3-propyl dinitrite (1,3-Pr(ONO)$_2$). Modeled OH$_{exp}$ values obtained from OFR369-i(iPrONO-$d_7$) and OFR369-i(iPrONO) (Fig. 2d) are shown for reference. Error bars for measurements represent $\pm 50\%$ uncertainty in OH$_{exp}$ and $I$-values.

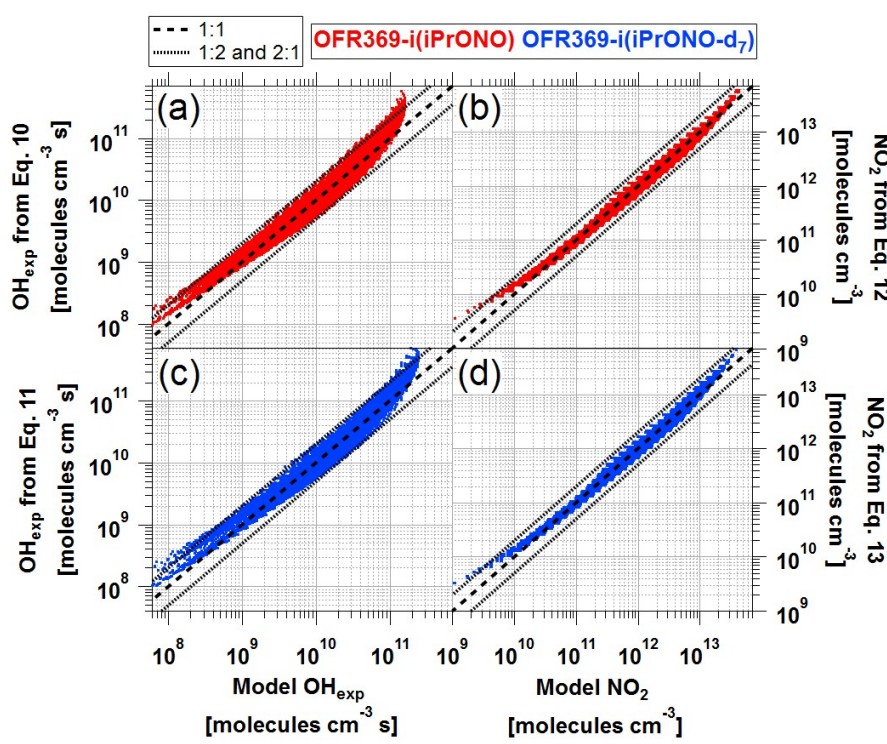

**Figure 5.** Comparision of $OH_{exp}$ and $NO_2$ values obtained from estimation equations and photochemical model for (a-b) OFR369-i(iPrONO) and (c-d) OFR369-i(iPrONO-d$_7$).

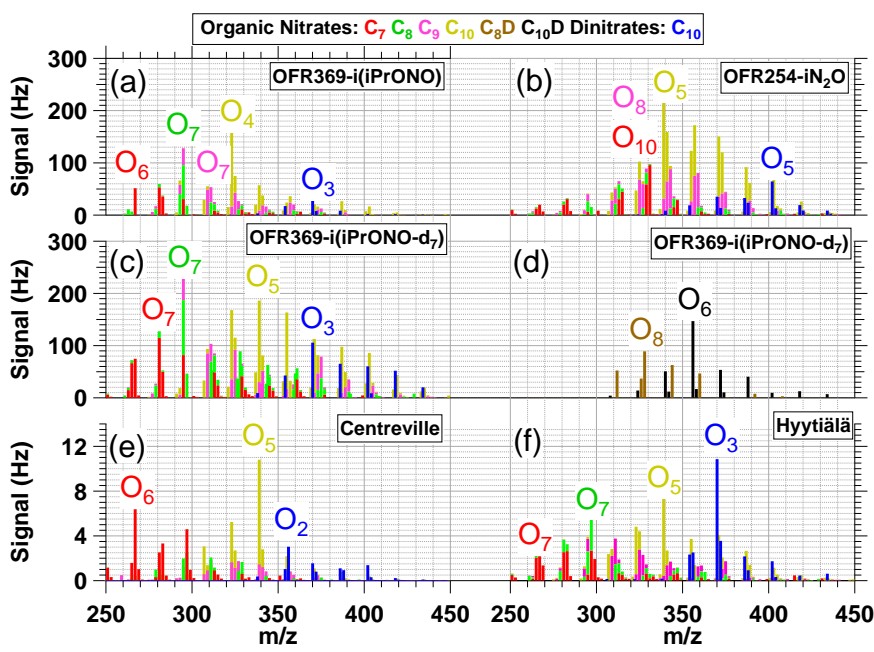

**Figure 6.** $NO_3^-$-CIMS spectra of nitrogen-containing $\alpha$-pinene photooxidation products with $C_{7-9}H_{9,11,13,15}NO_{5-10}$ ("$C_7$, $C_8$, $C_9$"), $C_{10}H_{15,17}NO_{4-14}$ ("$C_{10}$"), $C_8H_{8,10}DNO_{8-14}$ ("$C_8D$"), $C_{10}H_{14,16}DNO_{7-14}$ ("$C_{10}D$") or $C_{10}H_{16,18}N_2O_{6-13}$ ("$C_{10}$ dinitrate") formulas generated via (a) OFR369-i(iPrONO) (b) OFR254-iN$_2$O (H$_2$O = 1%, N$_2$O = 3.2%). (c,d) OFR369-i(iPrONO-d$_7$) and observed in ambient measurements at (e) Centreville, Alabama, United States (Massoli et al., 2018) (f) Hyytiala, Finland (Yan et al., 2016). "O$_x$" labels indicate number of oxygen atoms in corresponding signals (excluding 3 oxygen atoms per nitrate functional group).

**Table 1.** Absorption cross section ($\sigma_{A,\lambda}$; cm$^2$) or A + B bimolecular rate constant ($k_{A+B}$, cm$^3$ molec$^{-1}$ s$^{-1}$) reference values.

| $\sigma$ or $k$ | A | B | Value | Reference |
|---|---|---|---|---|
| $\sigma_{A,254}$ | iPrONO | – | $1.88 \times 10^{-18}$ | 1 |
| $\sigma_{A,350}$ | iPrONO | – | $1.11 \times 10^{-19}$ | 1,2 |
| $\sigma_{A,368}$ | iPrONO | – | $1.24 \times 10^{-19}$ | 1,2 |
| $\sigma_{A,254}$ | NO$_2$ | – | $1.05 \times 10^{-20}$ | 3 |
| $\sigma_{A,350}$ | NO$_2$ | – | $4.70 \times 10^{-19}$ | 3 |
| $\sigma_{A,368}$ | NO$_2$ | – | $5.60 \times 10^{-19}$ | 3 |
| k | iPrONO | OH | $7.20 \times 10^{-13}$ | 4 |
| k | iPrONO-d$_7$ | OH | $2.73 \times 10^{-13}$ | 5 |
| k | acetone | OH | $1.94 \times 10^{-13}$ | 6 |
| k | acetone-d$_6$ | OH | $3.21 \times 10^{-14}$ | 6 |
| k | CH$_3$CHO | OH | $1.5 \times 10^{-11}$ | 7 |
| k | CH$_3$C(O)O$_2$ | NO | $9 \times 10^{-12}$ | 8 |
| k | CH$_3$O$_2$ | NO | $7.7 \times 10^{-11}$ | 7 |
| k | HCHO | OH | $8.5 \times 10^{-12}$ | 7 |

[1]This work; [2]Raff and Finlayson-Pitts (2010); [3]Atkinson et al. (2004); [4]Raff and Finlayson-Pitts (2010); [5]Estimated from $k_{iPrONO+OH}$ scaled by relative rate constants of $n$-C$_3$H$_8$ + OH and $n$-C$_3$D$_8$ + OH (Nielsen et al., 1988, 1991); [6]Raff et al. (2005); [7]Burkholder et al. (2015); [8]Orlando and Tyndall (2012).