# Peer review of "HOx and NOx production in oxidation flow reactors via photolysis of isopropyl nitrite, isopropyl nitrite-d7, and 1,3-propyl dinitrite at $\lambda = 254$ , $350$ , and $369$ nm"

_Atmospheric Measurement Techniques, 2018_

## Referee Comment (RC1) · Anonymous Referee #1 · 11 Sep 2018

The author developed a new method using alkyl nitrite photolysis as a source of OH radical and NOx. Kinetic modeling was done to support that a much wider range of NO:HO2 ratio (10 -10000) was achieved. They present experimental and model characterization of the OH exposure and NOx levels generated via photolysis of C3 alkyl nitrites in the Potential Aerosol Mass (PAM) OFR. Together with chemical ionization mass spectrometer measurements of multifunctional oxidation, the author compared the products $\alpha$-pinene generated following the exposure of to HOx and NOx obtained using both isopropyl nitrite and O3 + H2O + N2O methods. This new method proposed

by Lambe et al. would open the prospect of OFR experiments at high NO. The paper is well written and organized. Few issues need to be addressed.

Major comments: While the author uses alkyl nitrates as a source of HOx and NOx in the oxidation flow reactor, their method provides a wider range of NO:HO2 ratio and lower OH exposure. The chemical ionization mass spectrometer measurements of $\alpha$-pinene oxidation products from different alkyl nitrates experiments are somehow comparable to some ambient measurement. While this method sounds promising, I would also be glad to know any disadvantage of using this method as it is important for the oxidation flow reactor users to avoid unwanted chemical reactions. For example, by photolysis of alkyl nitrate, we will generate a lot of RO, RO2 and R radicals. These radicals may also involve in the further reactions with intermediates from the oxidation of injected VOCs. Therefore produce additional products other than only from the oxidation of injected VOCs. I wonder if the author observes any such kind of products in their mass spectra data? Is this process significant?

Specific comments: 1) P1 Line 4: Delete "t" before "$\lambda$ = 254 nm" 2) P3 Line 18-22. The author tried to use a NOx analyzer (Model 405 nm, 2B Technologies) to quantify the NO/NO2 mixing ratio. As shown in Figure S1 (b), the alkyl nitrates also show absorption at 405 nm which is the working wavelength of the NOx analyzer. Though the absorption cross section of alkyl nitrates is about one order of magnitude lower than that of the NO2, the mixing ratio of alkyl nitrates can be much higher than NO2, thus bias the NO2 and NO measurement. To perform the measurement, the author needs to correct the absorption by the alkyl nitrates. 3) P4 Line 20-24. To test this hypothesize, the author can simply measure the emission spectra of the UV lamps. This measurement can provide a direct proof to see the influence of longer wavelengths emission lines. 4) A recent study by Ye et al. 2018 (ACP) found under wet conditions, heterogeneous uptake of SO2 onto organic aerosol was found to be the dominant sink of SO2, likely owing to reactions between SO2 and organic peroxides. This SO2 loss mechanism may bias the OH exposure measurement. 5) P6 Line 23-24. I suggest the author add

the reference data into that plot to show directly that their results are in good agreement with literature data. 6) P8 Line 6. Add "The model results showed that" before "For [iPrONO] $\leq$ 5 ppm". 7) P11 Line 28-29. How much can NO3 radical be produced in the OFR? If this is already included in the model, the author could show the results to indicate how important of NO3 radical oxidation.

Ye, J., Abbatt, J. P. D., and Chan, A. W. H.: Novel pathway of SO2 oxidation in the atmosphere: reactions with monoterpene ozonolysis intermediates and secondary organic aerosol, Atmos. Chem. Phys., 18, 5549-5565, hP6ttps://doi.org/10.5194/acp-18-5549-2018, 2018.

---

## Referee Comment (RC2) · Anonymous Referee #2 · 17 Oct 2018

**Review of Lambe et al. (2018)**

Summary and overall review: This manuscript evaluates the use of alkyl nitrite (AN) photolysis as an OH-precursor in an oxidation flow reactor (OFR). Experimental and model simulation approaches are used to constrain the parameters of interest to OFR studies such as the actinic flux calibration, amount of OH and $NO_x$ generation for different types of ANs as precursors. Empirical calibration equations are fit to observed data to create a domain of different OFR operational parameters such as residence time, external reactivity, etc. within which future AN-OFR experiments may operate. Finally, using chemical ionization mass spectrometry, it is shown that molecular structures of $\alpha$-pinene SOA formed in the AN-OFR bear resemblance to that of ambient SOA previously observed in terpene-rich environments.

The manuscript is topically relevant to AMT and builds on the body of literature regarding OFRs. However there are several shortcomings in the experimental description, outlined in my comments below, that must be addressed before it is ready for publication.

Major comment(s):
1. The manuscript would benefit from a clearer description of the conditions when a PAM/OFR user would want to deploy nitrite as the OH precursor instead of using OFR185, OFR254, or injecting HONO. This manuscript demonstrates that AN can be used as a HOx precursor, but putting this method into better context with existing OFR practices would improve the manuscript.
2. OH estimation from $SO_2$ and sulfate: What collection efficiency was assumed for sulfate particles in the ACSM? An example of the sulfur mass balance should be shown (e.g., $SO_2$ inlet, $SO_2$ that survives the OFR, particulate $SO_4$, $SO_2$ lost to walls or other surfaces), at least in the SI.
3. $OH_{exp}$ estimation in Section 2.2.2: This work achieves < 1 day of $OH_{exp}$ and thus the uncertainties with estimating $OH_{exp}$ warrant more attention. One of the earlier OFR studies by Lambe et al. (2011) accounted for the influence of humidity on the growth of $H_2SO_4$ particles upon $SO_2$ oxidation in the OFR. This section describes how calibration of $OH_{exp}$ v. particulate sulfate (from conventional OFR-254 method, hence in presence of humidity) was applied to measured particulate sulfate (from iPrONO photolysis, presumably also with humidity) to estimate $OH_{exp}$.
   - It would be beneficial to briefly discuss how humidity was controlled in both these experiments and whether or not it was accounted for in correction of ACSM-measured sulfate mass (unless sample was dried prior to ACSM sampling, in which case that should be specified).
   - It is not surprising that the sulfate mass responded linearly to increasing $[SO_{2,0}]$ in both these systems. The purpose of doing this inter-comparison was to see *how much* mass is formed in the iPrONO system v. in the conventional OFR-254 system, which would then imply how much $OH_{exp}$ is achieved in these two systems. Unless I am missing something, this comparison is not (but should be) plotted in Figure S5.
4. Page 6, L18-19: How were the reductions in quantum yields for R6 and R5 determined? This seems like a critical assumption in the modeling and it is not explained in much detail. What is the sensitivity of the model predictions to these quantum yields?

5. The presentation of the equations in Page 10 needs to be improved. First, there seems to be a formatting issue – the first equation appears as equations 3-6 and the second as equations 7-9. Each equation should have one number. Second, I don't understand where these equations came from. Where are the data these equations are fit to (it should at least be shown in the SI)? What is the quality of the fit? How was the functional form determined?
6. Section 3.5: The comparison between the OFR and ambient CIMS spectra are presented only as in-line text. This comparison would be more effective if done graphically.
7. Relevance of this study for "Mimicking polluted atmospheric conditions": the manuscript addresses a key limitation of the $N_2O$-OFR, in which, achieving $< 1$ equivalent day of $NO_x$-dependent SOA formation is challenging. While the use of ANs as OH (or OD) precursors is shown to be promising for achieving such low oxidative exposures in this study, this potentially makes OH suppression a major concern for *in-situ* deployment of the AN-OFR (Peng et al. 2015). The chemical composition of $\alpha$-pinene SOA formed in the AN-OFR (this study) bears resemblance to SOA previously observed in terpene-rich conditions in Centerville, Alabama and Hyytiälä, Finland (Yan et al., 2016; Massoli et al., 2018), suggesting that OH suppression may not be an issue. However, the manuscript lacks description of how much $\alpha$-pinene was injected into the OFR, whether OH suppression was a competing influence, and if yes, whether or not it was accounted for.

Minor comments:
- Abstract line 3: extra "t" before $\lambda$.
- Equation 1: I assume that density is for the liquid, but please specify.
- P1 L17: space needed before (Mao et al., 2009…). This error repeats several times in citations throughout the manuscript.
- P2 L13: in the presence of *humidified* air (if I am understanding the reactions correctly).
- Page 3, L14: the light manufacturer LCD Lighting is listed in this line but not the previous lines.
- P3 L18-23: Is it possible to include some numbers describing this interference (maybe in the SI)? How was the conclusion of "no avail" drawn? Did the 2B monitor read increasing $[NO_x]$ with increasing [iPrONO] injection into dark OFR? Since this AN photolysis is a unique aspect of this manuscript, I think instrumental caveats should be better described.
- Somewhere in the methods section, the authors should mention what was the flow through the OFR in the calibration experiments. The flow rate through OFR for CIMS experiments is mentioned later, but the flow rate in non-CIMS experiments is not mentioned anywhere.
- P8 L5: this sentence is confusing, because it suggests that measured values of $NO_x$ are shown in Figure 3, while in fact they are not. Should be reworded accordingly.
- P8 L30 (and Figure 4): the explanation of higher $NO_x$ offsetting OH production efficiency seems straightforward enough that it should be reproduced by KinSim. However, it seems the model was not run (or not plotted in Figure 4) for this OFR369-i(1,3-Pr(ONO)$_2$) scenario. Can this be explained?
- Again, the caption for Figure 4 is confusing because "measured and modeled values … of (iPrONO-d7) and (1,3-Pr(ONO)$_2$)" suggests that the modeled values for BOTH these precursors are plotted, while in fact the model was apparently not run for the latter precursor (this goes back to my previous comment).

- Sections 3.3.1 and 3.3.2: are these sub-sections relevant to their parent section 3.3? The parent section title only mentions (iPrONO-d7) and (1,3-Pr(ONO)$_2$). In fact, are these subsections even important enough to be placed in this part of the manuscript? There was no prior discussion of why MeONO and HFiPrONO are important OH precursors. These sub-sections abruptly build up the importance of these two precursors, and then rapidly declare that they are not suitable precursors in the OFR. The narrative flows smoother going directly from experimentally measuring OH$_{exp}$ to setting up estimation equations i.e., from P8 L33 directly to P9 L27. I suggest moving 3.3.1 and 3.3.2 to the end of the manuscript or to the SI.
- Sections 3.2 and 3.3 are really hitting the same hammer (how much OH$_{exp}$ is generated from precursor $X$) on different nails ($X$ = iPrONO, deuterated iPrONO, etc.). I don't see why they need to be separate sections.
- Figure S7b is missing a 1:1 line.
- P10 L18: NO2 needs a subscript.
- P11 L5: OFR operation details (flow rate, etc.) should be described in the Section 2.3. Also, amount of $\alpha$-pinene injected into OFR should be mentioned to give a sense of the OHR.
- P11 L11: compound nomenclature is missing some subscripts.
- P11 L20: this is a cool finding but does not readily jump out in Figure 5. I suggest adding a fourth panel showing a difference between the 5b and 5c (or 5a) spectra and zooming in the $m/z$ scale to show the just a few –OD containing sticks (e.g., from $m/z$ 310 to 360).
- Figure 5: There is enough empty space in each subfigure to include the dinitrite:nitrite ratio value. I suggest adding this in to quantify the "highest ratios observed in 5b" statement on P11 L24.
- Figure S7: units of OH$_{exp}$ are incorrect on both X- and Y-axes (s, not s$^{-1}$).

**References:**

Peng, Z.; Day, D. A.; Ortega, A. M.; Palm, B. B.; Hu, W. W.; Stark, H.; Li, R.; Tsigaridis, K.; Brune, W. H.; Jimenez, J. L. Non-OH chemistry in oxidation flow reactors for the study of atmospheric chemistry systematically examined by modeling. *Atmos. Chem. Phys. Discuss.* **2015**, *15* (17), 23543–23586.

Lambe, A. T.; Ahern, A. T.; Williams, L. R.; Slowik, J. G.; Wong, J. P. S.; Abbatt, J. P. D.; Brune, W. H.; Ng, N. L.; Wright, J. P.; Croasdale, D. R.; et al. Characterization of aerosol photooxidation flow reactors: Heterogeneous oxidation, secondary organic aerosol formation and cloud condensation nuclei activity measurements. *Atmos. Meas. Tech.* **2011**, *4* (3), 445–461.

Yan, C.; Nie, W.; Äijälä, M.; Rissanen, M. P.; Canagaratna, M. R.; Massoli, P.; Junninen, H.; Jokinen, T.; Sarnela, N.; Häme, S. A. K.; et al. Source characterization of highly oxidized multifunctional compounds in a boreal forest environment using positive matrix factorization. Atmos. Chem. Phys. **2016**, 16 (19), 12715–12731.

Massoli, P.; Stark, H.; Canagaratna, M. R.; Krechmer, J. E.; Xu, L.; Ng, N. L.; Mauldin, R. L.; Yan, C.; Kimmel, J.; Misztal, P. K.; et al. Ambient Measurements of Highly Oxidized Gas-Phase

Molecules during the Southern Oxidant and Aerosol Study (SOAS) **2013**. ACS Earth Sp. Chem. 2018, 2 (7), 653–672.

---

## Referee Comment (RC3) · Anonymous Referee #3 · 24 Oct 2018

Review of Lambe et al., AMTD, 2018

Summary and Recommendation

Authors introduce a new method for investigating NOx-dependent SOA formation pathways in oxidation flow reactors (OFRs). The new method uses alkyl nitrite photolysis to generate OH and NO2 with two different lights (254 nm and 369 nm). It is an improvement over previous methods used to study NOx-dependent SOA formation pathways in OFRs for three primary reasons. First, because it does not require extremely high

levels of ozone and it does not produce nitrate radical as a by-product; both ozone and nitrate radical also contribute to oxidation of SOA precursors and their presence in the reactor creates major challenges for deconvolving contributions from the different oxidants. Second, it can be run with 369 nm lamps and avoid photolytic losses of SOA precursors that can occur with the more commonly used 254 nm lamps. Third, unlike batch reaction chamber studies which can only probe over timescales of hours to $\sim$1 day, the OFR can be used to probe oxidative aging equivalent to multiple days.

There were a number of challenges using the new method. First, the alkyl nitrites presented an interference in the NOx analyzer, and they had to use a photochemical model to estimate NOx in lieu of a direct measurement. Second, the NOx generated from alkyl nitrite photolysis introduced an interference in the SO2 analyzer and made it difficult to determine OH exposure. They corrected for this by performing an offline calibration relating SO2 decay and particulate sulfate to OH exposure.

The technique does not achieve equivalent OH exposures longer than one day, but does appear to be a promising technique for OFR users to study high NOx SOA chemistry. This is particularly true for oxidation conditions using 369 nm lights. I recommend the manuscript for publication after the following comments are addressed.

Major Comments

Elaborate on alkyl nitrite interference with NOx analyzer. Text says they attempted to correct for the interference "to no avail" (p.3, l. 22). What was the issue that prevented this correction?

OH exposure calibration: not entirely clear how this calibration provided a measure of equivalent OH exposure from OH and NO2 generated from alkyl nitrites. It could be cleared up by better describing the link between Figure S4 and Figure S5. The connection is lost by lack of clarity regarding the x-axis in Figure S4 and explicitly stating at the end of the paragraph how the relationship in Figure S5 is used to estimate OH exposure as presented in Figure S4. In Figure S4, does the x-axis "sulfate" refers to both

SO2 decay in the gas-phase and sulfate measured in the particles with the ACSM? There is something missing in the description that connects how the researchers propose to then presumably use the ACSM sulfate measurement in the presence of alkyl nitrites to estimate initial SO2 (Figure S5) and then relate that back to OH exposure using the relationship shown in Figure S4.

Section 2.4: Additional reactions included in the model and all input parameters, rate constants, etc. are stated very clearly. Thank you. Uncertainties for actinic flux and organic nitrite concentration were also discussed very clearly. This model is being used to estimate NO and NO2 because of the NOx analyzer interference from alkyl nitrites (Section 2.2). I did not see an estimate of the uncertainty for NO and NO2 estimates. Please add those.

Figure 2: The explanation for lower OHexp values with 254 nm lights versus 350 nm and 369 nm was not very clear (page 7-8). In particular, elaborate on the concept, "Because $\sigma$iPrONO;369 « $\sigma$iPrONO;254 (Table 1), the effect of photolysis wavelength on [NO2] is proportional to $\sigma$iPrONO, as expected" and how that relates to reduced OH exposure at 254 nm. The other explanation for reduced OHexp with 254 nm lights (decomposition of iC3H7O radical) is described in adequate detail, but it would help to better clarify the proportion of iPrONO that decomposes at this wavelength to provide more context for how significant this pathway is at the shorter wavelengths.

Section 3.4: OHexp and NO2 estimation equations If this section is going to be included in the main text of the results, the Figure S7 should also be included in the main text since that figure summarizes the main results from this section. Alternatively, the entire section could be moved to supplement. Can you clarify the significance of the results from this section somewhere in the text?

One of the goals was to identify the optimal range of conditions for using the new alkyl nitrite method. Can you state the recommended "optimal" conditions more clearly AND also put the OFR conditions within that range into context relative to atmospheric

conditions, particularly for NO:NO2 ratios and RO2:HOx? Are there certain conditions that should particularly be avoided? Can you make those more clear as well?

One of the stated benefits of OFRs in the intro is their ability to simulate "multiple days of equivalent atmospheric exposure." (P. 2, L. 4). Looking at Figures 2-4, it does not appear the OFR was capable of obtaining more than 1 day equivalent OH exposure either using the alkyl nitrite technique. Can you comment on how this timescale compares with smog chamber experiments investigating similar NOx pathways?

It would be helpful to see a direct comparison between OH exposure using the alkyl nitrite method versus other techniques that have been used to probe the high NOx pathway (for example, N2O addition).

Minor Comments

P. 2, L. 28-30: alkyl nitrite were stored in amber vials and refrigerated until use. Can you clarify how long they can be stored and approximately how much time passed from synthesis to experimental use in these experiments?

---

## Author Comment (AC1) · 15 Dec 2018

**Response to reviewers for the paper "HO$_x$ and NO$_x$ production in oxidation flow reactors via photolysis of isopropyl nitrite, isopropyl nitrite-d$_7$, and 1,3-propyl dinitrite at λ = 254, 350, and 369 nm."**

We thank the reviewers for their comments on our paper. To guide the review process we have copied the reviewer comments in black text. Our responses are in regular blue font. We have responded to all the referee comments and made alterations to our paper (**in bold text**).

**Anonymous Referee #1**
The author developed a new method using alkyl nitrite photolysis as a source of OH radical and NOx. Kinetic modeling was done to support that a much wider range of NO:HO2 ratio (10 -10000) was achieved. They present experimental and model characterization of the OH exposure and NOx levels generated via photolysis of C3 alkyl nitrites in the Potential Aerosol Mass (PAM) OFR. Together with chemical ionization mass spectrometer measurements of multifunctional oxidation, the author compared the products α-pinene generated following the exposure of to HOx and NOx obtained using both isopropyl nitrite and O3 + H2O + N2O methods. This new method proposed by Lambe et al. would open the prospect of OFR experiments at high NO. The paper is well written and organized. Few issues need to be addressed.

R1.1) While the author uses alkyl nitrates as a source of HOx and NOx in the oxidation flow reactor, their method provides a wider range of NO:HO2 ratio and lower OH exposure. The chemical ionization mass spectrometer measurements of α-pinene oxidation products from different alkyl nitrates experiments are somehow comparable to some ambient measurement. While this method sounds promising, I would also be glad to know any disadvantage of using this method as it is important for the oxidation flow reactor users to avoid unwanted chemical reactions. For example, by photolysis of alkyl nitrate, we will generate a lot of RO, RO2 and R radicals. These radicals may also involve in the further reactions with intermediates from the oxidation of injected VOCs. Therefore produce additional products other than only from the oxidation of injected VOCs. I wonder if the author observes any such kind of products in their mass spectra data? Is this process significant?

We modified the text as follows:

P12, L18-26: "Taken together, OFR254/**OFR**185-iN$_2$O and OFR369-i(iPrONO/iPrONO-d$_7$) are complementary methods that provide additional flexibility for NO$_x$-dependent OFR studies. **OFR254/OFR185-iN$_2$O generate variable-NO$_x$ photooxidation conditions (NO:HO2≈0 - 100), and are suitable for the characterization of multigenerational oxidative aging processes at up to OH$_{exp}$ ~ (5-10)*10$^{11}$ molecules cm$^{-3}$ s (~5-10 eq. days).** OFR369-i(iPrONO)/OFR369-i(iPrONO-d$_7$) generate high-NO photooxidation conditions (NO:HO$_2$≈10 - 10000) with minimal O$_3$ and NO$_3$ formation at longer photolysis wavelength than OFR254/185-iN$_2$O. We anticipate that alkyl nitrite photolysis is **advantageous** for the characterization of first-generation, high-NO$_x$ photooxidation products of most precursors **at up to OH$_{exp}$ ~ 1*10$^{11}$ molecules cm$^{-3}$ s (1 eq. day), which is comparable to environmental chambers investigating high-NO$_x$ conditions.** The generation of OD (rather than OH) via OFR369-i(iPrONO-d$_7$) may be useful in photooxidation

studies of unsaturated precursors due to the shift on the m/z of the addition products, though at the potential expense of generating more complex distributions of oxidation products. **Potential disadvantages of using alkyl nitrite photolysis as a $HO_x$ source are: (1) restriction to high-NO photochemical conditions; (2) restriction to $OH_{exp}$ of 1 eq. day or less; (3) additional complexity involved with integration of the alkyl nitrite source (compared to $O_3$ + $H_2O$ + $N_2O$); (4) potential inability to retrofit a specific OFR design with blacklights; (5) it acts as an interference that precludes NOx measurements by chemiluminescence detection.**

In regards to the reviewer's comment about R, RO, and RO2 radicals produced from isopropyl nitrite photolysis, the species that are treated in our model (R5-R17) include:

R: $CH_3$
RO: $i\text{-}C_3H_7O$, $CH_3CO$, HCO
RO2: $CH_3O_2$, $CH_3C(O)O_2$

Of the above species, in the presence of oxygen -- typically the case in most modern OFR studies -- all of the R and RO species ($CH_3$, $i\text{-}C_3H_7O$, $CH_3CO$, and HCO) are too short-lived to directly participate in reactions with $RO_2$ radicals formed from the oxidation of injected VOCs:

- $CH_3$ + $O_2$ generates $CH_3O_2$
- $i\text{-}C_3H_7O$ + $O_2$ mostly generates $HO_2$ and acetone
- $CH_3CO$ + $O_2$ generates $CH_3C(O)O_2$
- HCO + $O_2$ generates CO + $HO_2$

Thus, the most potentially problematic species include $CH_3O_2$ and $CH_3C(O)O_2$, which could participate in reactions with organic peroxy radicals generated from photooxidation of injected VOCs. Because generation of $CH_3O_2$ and $CH_3C(O)O_2$ only proceeds via iPrONO + hv → $CH_3CHO$ + $CH_3$• + NO  (R6), which has an estimated quantum yield of ~0.04 (P6, L16), the relative importance of these reactions is likely minor.

We modified the text as follows:

P6, L15-L16: "We assumed the quantum yield of Reaction R5 to be 0.5 **above 350 nm** (Raff and Finlayson-Pitts, 2010). **We assumed the quantum yield** of Reaction R6 to be 0.04 above 350 nm (value for t-butyl nitrite) (Calvert and Pitts, 1966), **suggesting minimal influence of $CH_3O_2$ and $CH_3C(O)O_2$ under these conditions that are generated via Reactions R7, R10, and R11 following iPrONO decomposition to $CH_3$ and $CH_3CHO$ via Reaction R6. At 254 nm, the influence of $CH_3O_2$ and $CH_3C(O)O_2$ on ensuing photochemistry may be more significant. This is due to a higher quantum yield of Reaction R6 at 254 nm, which is estimated to be 0.86 under vacuum (Calvert and Pitts, 1966)."**

R1.2) P1 Line 4: Delete "t" before "λ = 254 nm"

Deleted.

R1.3) P3 Line 18-22: The author tried to use a NOx analyzer (Model 405 nm, 2B Technologies) to quantify the NO/NO2 mixing ratio. As shown in Figure S1 (b), the alkyl nitrates also show absorption at 405 nm which is the working wavelength of the NOx analyzer. Though the absorption cross section of alkyl nitrates is about one order of magnitude lower than that of the NO2, the mixing ratio of alkyl nitrates can be much higher than NO2, thus bias the NO2 and NO measurement. To perform the measurement, the author needs to correct the absorption by the alkyl nitrates.

Please see the text on P3, L23, where we stated that "we constrained [NO] and [$NO_2$] using the photochemical model discussed in Section 2.4" because we had difficulty correcting for absorption by the alkyl nitrites.

R1.4) P4 Line 20-24: To test this hypothesize, the author can simply measure the emission spectra of the UV lamps. This measurement can provide a direct proof to see the influence of longer wavelengths emission lines.

This is a fair point. We did not have access to an instrument that could measure the emission spectra of the 254 nm UV lamps; in the end, because OFR254-i(iPrONO) is not recommended, we did not pursue it further.

R1.5): A recent study by Ye et al. 2018 (ACP) found under wet conditions, heterogeneous uptake of SO2 onto organic aerosol was found to be the dominant sink of SO2, likely owing to reactions between SO2 and organic peroxides. This SO2 loss mechanism may bias the OH exposure measurement.

Thank you for the reference. In this work, OH exposure measurements were not conducted in the presence of organic aerosol (no VOCs were injected aside from alkyl nitrites, which themselves do not generate aerosol). Therefore we think that this is not an issue that is relevant to our results. However, we added the following sentence to the end of Section 2.2.2 to alert readers of the potential effect:

**"While not applicable in this work, we note that heterogeneous uptake of $SO_2$ onto organic aerosol may bias OH exposure measurements (Ye et al., 2018)."**

We added the following citation to references:

**Ye, J., Abbatt, J. P. D., and Chan, A. W. H.: Novel pathway of $SO_2$ oxidation in the atmosphere: reactions with monoterpene ozonolysis intermediates and secondary organic aerosol, Atmos. Chem. Phys., 18, 5549-5565, https://doi.org/10.5194/acp-18-5549-2018, 2018.**

R1.6) P6 Line 23-24: I suggest the author add the reference data into that plot to show directly that their results are in good agreement with literature data.

We added isopropyl nitrite absorption cross sections obtained from $\lambda = 300$ to 450 nm by Raff and Finlayson-Pitts (2010) (black dashed line)  to a revised Figure S1 shown below:

[Figure]

R1.7) P8 Line 6: Add "The model results showed that" before "For [iPrONO] ≤ 5 ppm".

We modified the text as follows:

P8, L6: "Figure 3 shows measured and modeled $OH_{exp}$ and NOx concentrations obtained from photolysis of 0.5 to 20 ppm iPrONO [...] **The model results showed that for** [iPrONO] ≤5 ppm, $OH_{exp}$ increased with increasing [iPrONO] because the rate of OH production increased faster than the rate of OH destruction from reaction with iPrONO and $NO_2$. For [iPrONO] > 5 ppm, the opposite was true and $OH_{exp}$ plateaued or decreased. A maximum $OH_{exp}$ =$7.8 \times 10^{10}$ molecules $cm^{-3}$ s was achieved via photolysis of 10ppm iPrONO, with corresponding modeled [NO] and [$NO_2$] values of 148 and 405 ppb respectively."

R1.8) P11 Line 28-29: How much can NO3 radical be produced in the OFR? If this is already included in the model, the author could show the results to indicate how important of NO3 radical oxidation.

The maximum $NO_3$ concentration in the model cases in this study is only ~1 ppt, since there is no $O_3$ in OFR-i(iPrONO) and the second step of the $NO_2 \rightarrow HNO_3 \rightarrow NO_3$ oxidation chain by OH is slow. We thus do not report the negligible $NO_3$ concentrations in the figures but added the following sentence at the end of Section 3.2:

**"Modeled $NO_3$ concentrations were negligible in OFR-i(iPrONO) (<~1 ppt) because there is no $O_3$ present and $NO_3$ production via $NO_2$ + OH →$HNO_3$ and $HNO_3$ + OH →$NO_3$ + $H_2O$ reactions was small."**

---

## Author Comment (AC2) · 15 Dec 2018

**Response to reviewers for the paper "HO$_x$ and NO$_x$ production in oxidation flow reactors via photolysis of isopropyl nitrite, isopropyl nitrite-d$_7$, and 1,3-propyl dinitrite at λ = 254, 350, and 369 nm."**

We thank the reviewers for their comments on our paper. To guide the review process we have copied the reviewer comments in black text. Our responses are in regular blue font. We have responded to all the referee comments and made alterations to our paper (**in bold text**).

**Anonymous Referee #2**

Summary and overall review: This manuscript evaluates the use of alkyl nitrite (AN) photolysis as an OH-precursor in an oxidation flow reactor (OFR). Experimental and model simulation approaches are used to constrain the parameters of interest to OFR studies such as the actinic flux calibration, amount of OH and NOx generation for different types of ANs as precursors. Empirical calibration equations are fit to observed data to create a domain of different OFR operational parameters such as residence time, external reactivity, etc. within which future AN-OFR experiments may operate. Finally, using chemical ionization mass spectrometry, it is shown that molecular structures of α-pinene SOA formed in the AN-OFR bear resemblance to that of ambient SOA previously observed in terpene-rich environments. The manuscript is topically relevant to AMT and builds on the body of literature regarding OFRs. However there are several shortcomings in the experimental description, outlined in my comments below, that must be addressed before it is ready for publication.

R2.1): The manuscript would benefit from a clearer description of the conditions when a PAM/OFR user would want to deploy nitrite as the OH precursor instead of using OFR185, OFR254, or injecting HONO. This manuscript demonstrates that AN can be used as a HOx precursor, but putting this method into better context with existing OFR practices would improve the manuscript.

Please see our response and updates to the paper text in response to a similar comment R.1.1 regarding comparison of OFR369-i(iPrONO) and OFR185/OFR254-iN2O. We anticipate that HONO will not be a useful HOx precursor in OFRs, as discussed in a new subsection below (please note that section has changed from Section 3.3.x to Section 3.5.x in response to comment 2.18):

**3.5.2 Nitrous acid (HONO)**
**HONO is also commonly used as an OH radical source in environmental chamber studies. To evaluate its potential application in OFRs, we examined previous measurements in an environmental chamber equipped with blacklights, where photolysis of 3-20 ppm HONO generated initial [OH]~6×10$^7$ molecules cm$^{-3}$ (Cox et al., 1980) which is 3.3 times lower than [OH] obtained from comparable levels of MeONO (Section 3.5.1). Lower OH$_{exp}$ achieved from HONO photolysis is presumably due to higher OH reactivity of HONO relative to MeONO/iPrONO. Additionally, HONO is difficult to prepare without NO$_2$ impurities (Febo et al., 1995) that may cause additional OH suppression. For these reasons, we believe that**

**there is no advantage to using HONO as a $HO_x$ precursor in OFRs.**

We have added the following references:

**A. Febo, C. Perrino, M. Gherardi, and R. Sparapani. Evaluation of a High-Purity and High-Stability Continuous Generation System for Nitrous Acid. Environmental Science & Technology 1995 *29* (9), 2390-2395.DOI: 10.1021/es00009a035.**

**Richard A. Cox, Richard G. Derwent, and Michael R. Williams. Atmospheric photooxidation reactions. Rates, reactivity, and mechanism for reaction of organic compounds with hydroxyl radicals Environmental Science & Technology 1980 *14* (1), 57-61. DOI: 10.1021/es60161a007**

R2.2) OH estimation from SO2 and sulfate: (i) What collection efficiency was assumed for sulfate particles in the ACSM? (ii) An example of the sulfur mass balance should be shown (e.g., SO2 inlet, SO2 that survives the OFR, particulate SO4, SO2 lost to walls or other surfaces), at least in the SI.

(i) We assumed CE = 1, but for our purpose, the absolute CE value doesn't matter provided that the CE of sulfuric acid particles generated by $SO_2$ + OH via conventional OFR254 or via alkyl nitrite photolysis is the same. This assumption is justified based on the fact the humidity was similar for OFR254 and alkyl nitrite experiments and no ammonia (aside from presumably trace background levels) were present.

We modified the text as follows:

P4-5,L31-2: " To relate the measured $[SO_{2,0}]$ and sulfate to OHexp, we conducted an offline calibration where 493 ppb SO2 was added to the reactor and OH was generated via $O_3$ + hv254 $\rightarrow O(^1D)$ + $O_2$ followed by $O(^1D)$ + $H_2O \rightarrow 2OH$ in the absence of $NO_x$. The reactor was operated at the same residence time **and humidity** used in alkyl nitrite experiments, **although we note that humidity will not change the response of the ACSM to sulfuric acid aerosols**. **Because no particulate ammonia was present aside from trace background levels, we assumed an ACSM collection efficiency of unity for the sulfate particles.**"

(ii) A sulfur mass balance is not possible because we could not unambiguously measure the $SO_2$ that survives the OFR due to apparent interferences in the $SO_2$ measurement (P4, L28). We added a new supplemental figure that illustrates this:

[Figure]

**Figure S5**. Example time series of SO$_2$ mixing ratio and irradiance (UV intensity) measured during a representative OFR369-i(iPrONO) OH$_{exp}$ calibration. (A) Began SO$_2$ addition at OFR inlet with lamps off; 9.3 ppm iPrONO also added

R2.3) OHexp estimation in Section 2.2.2: This work achieves < 1 day of OHexp and thus the uncertainties with estimating OHexp warrant more attention. One of the earlier OFR studies by Lambe et al. (2011) accounted for the influence of humidity on the growth of H2SO4 particles upon SO2 oxidation in the OFR. This section describes how calibration of OHexp v. particulate sulfate (from conventional OFR-254 method, hence in presence of humidity) was applied to measured particulate sulfate (from iPrONO photolysis, presumably also with humidity) to estimate OHexp.

R2.3a): It would be beneficial to briefly discuss how humidity was controlled in both these experiments and whether or not it was accounted for in correction of ACSM measured sulfate mass (unless sample was dried prior to ACSM sampling, in which case that should be specified).

We modified the text as follows:

P3, L11: "The relative humidity (RH) in the reactor was controlled in the range of 31-63% at 21-32°C **using a Nafion humidifier (Perma Pure LLC), with corresponding** H$_2$O volumetric mixing ratios of approximately 1.5-1.7%.

Please also see our response to R2.2, where we note that humidity does not affect the ACSM response to sulfuric acid aerosols.

R2.3b): It is not surprising that the sulfate mass responded linearly to increasing [SO2,0] in both these systems. The purpose of doing this inter-comparison was to see how much mass is formed

in the iPrONO system v. in the conventional OFR-254 system, which would then imply how much OHexp is achieved in these two systems. Unless I am missing something, this comparison is not (but should be) plotted in Figure S5.

It was necessary to demonstrate a linear response between sulfate mass and [SO2,0] to illustrate that the sulfate particles were efficiently transmitted through the ACSM inlet aerodynamic lens (P5, L6-7). If they were not (e.g. too small or too large vs. the lens transmission window), we anticipate that the response would have been nonlinear.

We have revised Figure S4 (below; now Figure S6 in revised manuscript) to include the sulfate mass measured following $SO_2$ oxidation in the alkyl nitrite photolysis experiments compared with OFR254 experiments. The corresponding OH exposure for the alkyl nitrite systems was obtained by extrapolating the OFR254 calibration data to lower OH exposure.

[Figure]

Figure S6. Calibrated OHexp obtained following reaction of 493 ppb $SO_2$ with OH generated via $O_3$+ $hv_{254}\rightarrow O(^1D)+O_2$ followed by $O(^1D)+H_2O\rightarrow 2OH$ in the absence of $NO_x$ **(red symbols)**. **The calibration equation was applied to measurements of sulfate formed during** alkyl nitrite **photolysis** experiments **(blue symbols) where $SO_2$ was added at the**

R2.4) Page 6, L18-19: How were the reductions in quantum yields for R6 and R5 determined? This seems like a critical assumption in the modeling and it is not explained in much detail. What is the sensitivity of the model predictions to these quantum yields?

We modified the text between L17-19 to clarify our rationales of this assumption. We also decide to change the upper limit quantum yield for Reaction R5 at 254 nm from 0.40 to 0.50 to reflect the value obtained by Raff and Finlayson-Pitts above 350 nm wavelength. The text now reads:

"At 254 nm, Calvert and Pitts (1966) estimated the quantum yield of Reaction R6 to be 0.86 under vacuum. **Assuming that all 254 nm photons initiate photolysis, the corresponding quantum yield of Reaction R5 is 0.14. Due to** collisional deactivation **at 1 atm** that prevent**s** i-$C_3H_7O\bullet$ decomposition**, the quantum yield of Reaction R5 at λ= 254 nm and 1 atm is expected to be higher than 0.14. Because quantum yield measurements were unavailable at these conditions, we applied an upper limit quantum yield of 0.50 as applicable at λ>350 nm and 1 atm (Raff and Finlayson-Pitts, 2010). We calculated a corresponding nominal quantum yield of 0.32 by averaging the lower and upper limit values of 0.14 and 0.50, resulting in a quantum yield of 0.68 for Reaction R6.**"

Regarding the sensitivity of the model predictions to the quantum yield of Reaction R6, we modified text to Page 7, L29 to read:

"**Higher $NO_2$ concentrations were modeled at λ = 254 nm than at λ = 369 nm because more iPrONO was photolyzed and the $NO_2$ yield was only weakly dependent on the fate of i-C3H7O•. For example, NO is converted to NO2 either via reaction with HO2 obtained via Reaction R5 or CH3O2• and CH3C(O)O2• obtained via Reaction R6.** However, the effect of photolysis wavelength on **NO and** $OH_{exp}$ was different. Specifically, the highest NO concentration and $OH_{exp}$ was achieved via OFR369-i(iPrONO). $OH_{exp}$ achieved via OFR369-i(iPrONO) was slightly higher than $OH_{exp}$ attained using OFR350-i(iPrONO), likely because photolysis of both iPrONO and $NO_2$, whose reaction with OH suppresses $OH_{exp}$, is more efficient at λ = 369 nm than at λ = 350 nm (Figure S1 and Table 1). Further, the NO and OH yields achieved via OFR254-i(iPrONO) were suppressed due to significant (>**68**%) decomposition of i-$C_3H_7O$ (Calvert and Pitts, 1966). The products of i-$C_3H_7O$ decomposition, i.e., $CH_3CHO$ and $CH_3\bullet$, both have adverse effects with regard to our experimental goals: $CH_3CHO$ is reactive toward OH and can thus suppress OH; the $RO_2\bullet$ formed through this reaction, $CH_3C(O)O_2\bullet$, consumes NO **and generates $NO_2$** but does not generate OH; $CH_3\bullet$ rapidly converts to $CH_3O_2\bullet$, which also consumes NO **and generates $NO_2$** but does not directly produce OH. **The dependence of OH, NO and $NO_2$ on the quantum yields of Reactions R5 and R6 was confirmed by sensitivity analysis of uncertainty propagation inputs and outputs as described in Section 2.4. $OH_{exp}$ and NO were strongly anticorrelated with the quantum yield of Reaction R6, whereas the correlation between NO2 and the quantum yield of Reaction R6 was negligible.**"

R2.5): The presentation of the equations in Page 10 needs to be improved. First, there seems to be a formatting issue – the first equation appears as equations 3-6 and the second as equations 7-9. Each equation should have one number. Second, I don't understand where these equations came from. Where are the data these equations are fit to (it should at least be shown in the SI)? What is the quality of the fit? How was the functional form determined?

The equation formatting issue appears to be related to our attempt to implement multi-line equations using the Copernicus LaTeX template. We will follow up with the copy editing staff to resolve this issue.

To address the other questions from the reviewer, we modified text to Page 10, L11 to read:

"Fit coefficients were obtained by fitting Equations 3 and 6 to $OH_{exp}$ model results over the following range of OFR parameters: ([iPrONO/iPrONO-d$_7$]; 0.2-20 ppm), $I_{369}$ ($1\times10^{15}$ - $2\times10^{16}$ photons cm$^{-2}$ s$^{-1}$), $OHR_{ext}$ (1-200 s$^{-1}$), and residence time, τ, between 30 and 200 sec. **We explored 11 logarithmically evenly distributed values in these ranges for each parameter, and thus performed simulations for 14641 model cases in total. To determine the functional form of Eqs. (3) and (6), we used the sum of the logarithms of first-, second-, and third-order terms of the four parameters and iteratively removed the terms with very small fit coefficients until further removal of remaining terms significantly worsened the fit quality."**

We also modified text on Page 10, L20 to read:

"Thus, we derived $NO_2$ estimation equations for OFR369-i(iPrONO) (Eq. 10) and OFR369-i(iPrONO-d$_7$) (Eq. 11) as a function of [RONO], $I_{369}$, and τ**, to all of which NO$_2$ is roughly proportional,** over the same phase space of model results used to **fit** Eqs. 3 and 6:"

The output data points of the model runs for fitting estimation equations (and the corresponding quantities estimated by the fitted equations) had already been shown in Fig. S7 of the AMTD version. The mean absolute values of the relative deviations of the equation estimates from the model outputs had already been reported to be 29% and 19% in Page 10, L14 and 26 for $OH_{exp}$ and $NO_2$, respectively.

R2.6) Section 3.5: The comparison between the OFR and ambient CIMS spectra are presented only as in-line text. This comparison would be more effective if done graphically.

To Figure 5 (Figure 6 in the revised manuscript) we added panels (e) and (f) containing ambient $NO_3^-$-CIMS spectra obtained from high-$NO_x$ photochemical conditions in Centreville, Alabama, USA (Massoli et al., 2018) and in Hyytiala, Finland (Yan et al., 2016) We implemented an additional suggestion by this reviewer to add a separate panel (d) showing the -OD containing sticks (see R2.24).

[Figure]

**Figure 6**. $NO_3^-$-CIMS spectra of nitrogen-containing α-pinene photooxidation products with $C_{7-9}H_{9,11,13,15}NO_{5-10}$ ("$C_7$, $C_8$, $C_9$"), $C_{10}H_{15,17}NO_{4-14}$ ("$C_{10}$"), $C_8H_{8,10}DNO_{8-14}$ ("$C_8D$"), $C_{10}H_{14,16}DNO_{7-14}$ ("$C_{10}D$") or $C_{10}H_{16,18}N_2O_{6-13}$ ("$C_{10}$ dinitrate") formulas generated via (a) OFR369-i(iPrONO) (b) OFR254-iN$_2$O (H$_2$O = 1%, N$_2$O = 3.2%). (c,**d**) OFR369-i(iPrONO-d7) **and observed in ambient measurements at (e) Centreville, Alabama, United States (Massoli et al., 2018) (f) Hyytiala, Finland (Yan et al., 2016).** "Ox" labels indicate number of oxygen atoms in corresponding signals (excluding 3 oxygen atoms per nitrate

We modified the text as follows:

"The ability of OFR369-i(iPrONO) and OFR369-i(iPrONO-d7) to mimic polluted atmospheric conditions can be evaluated by comparing signals observed in Figure **6** with published $NO_3^-$ - CIMS spectra obtained in Centreville, AL, USA (Massoli et al., 2018) and in Hyytiala, Finland (Yan et al., 2016). Both measurement locations are influenced by local biogenic emissions mixed with occasional anthropogenic outflow. **Figures 6e and 6f were obtained on 25 June 2013 (7:30–11:00 Centreville time) and 11 April 2012 (10:00-13:00 Hyytiala time) respectively. The mean NO mixing ratios during these periods were 0.53 ± 0.17 (Centreville) and 0.27 ± 0.09 ppb (Hyytiala).** In Centreville, the largest C10 nitrate and dinitrate species were **$C_{10}H_{15}NO_8$ and $C_{10}H_{16}N_2O_8$; in Hyytiala, $C_{10}H_{15}NO_8$ and $C_{10}H_{16}N_2O_9$ were the largest C10 nitrate/dinitrate signals. Elevated $C_{10}$ dinitrate levels during the daytime in Hyytiaila (Figure 6f) suggests their formation from monoterpenes via two OH reactions followed by two RO$_2$ + NO termination reactions, as proposed earlier. Overall, Figure 6 shows that many of the C7-C10 nitrogen-containing compounds observed in Centreville and Hyytiala** were generated via OFR369-i(iPrONO), OFR369-i(iPrONO-d7) and OFR254-iN$_2$O. **D**ue to the local nature of the ambient terpene emissions at the Centreville and Hyytiala sites, the associated photochemical

age was presumably <1 day. Thus, while the ambient $NO_3^-$-CIMS spectra at those sites were more complex and contained contributions from precursors other than α-pinene, the oxidation state of the ambient terpene-derived organic nitrates was more closely simulated via OFR369-i(iPrONO) or OFR369-i(iPrONO-d7), where the largest C10 nitrates and dinitrates were $C_{10}H_{15}NO_7$ and $C_{10}H_{16}N_2O_9$ (OFR369-i(iPrONO); Figure 5a), and $C_{10}H_{15}NO_8$, $C_{10}H_{15}NO_9$ and $C_{10}H_{16}N_2O_9$ (OFR369-i(iPrONOd7); Figure 5c). By comparison, $C_{10}H_{15}NO_8$ and $C_{10}H_{16}N_2O_{11}$ were the largest nitrate and dinitrate species generated via OFR254-iN2O (Figure 5b)."

R2.7) Relevance of this study for "Mimicking polluted atmospheric conditions": the manuscript addresses a key limitation of the N2O-OFR, in which, achieving < 1 equivalent day of NOx dependent SOA formation is challenging. While the use of ANs as OH (or OD) precursors is shown to be promising for achieving such low oxidative exposures in this study, this potentially makes OH suppression a major concern for in-situ deployment of the AN-OFR (Peng et al. 2015). The chemical composition of α-pinene SOA formed in the AN-OFR (this study) bears resemblance to SOA previously observed in terpene-rich conditions in Centerville, Alabama and Hyytiälä, Finland (Yan et al., 2016; Massoli et al., 2018), suggesting that OH suppression may not be an issue. However, the manuscript lacks description of how much α-pinene was injected into the OFR, whether OH suppression was a competing influence, and if yes, whether or not it was accounted for.

We modified the text as follows:

P11, L3: "To evaluate the efficacy of OFR369-i(iPrONO), OFR369-i(iPrONO-d$_7$), and OFR254-iN$_2$O [...] the reactor was operated with a residence time of approximately 80 sec to accommodate the undiluted $NO_3^-$-CIMS inlet flow requirement (10.5 L min$^{-1}$). OFR369-i(iPrONO) and OFR369-i(iPrONO-d$_7$) were operated using $I_{369}$ = 6.5×10$^{15}$ photons cm$^{-2}$ s$^{-1}$, >7 ppm nitrite, **and 500 ppb a-pinene**. OFR254-iN$_2$O was operated using $I_{254}$ =3.2×10$^{15}$ photons cm$^{-2}$ s$^{-1}$, 5 ppm O$_3$ + 1% H$_2$O + 3.2% N$_2$O, **and 16 ppb α-pinene**. Corresponding **calculated** OH exposures were 2.9×10$^{10}$, 5.9×10$^{10}$ and **5.0×**10$^{11}$ molecules cm$^{-3}$ s, respectively, **in the absence of OH consumption due to α-pinene. These calculated steady-state OH$_{exp}$ values decreased to 8.5×10$^8$, 6.8×10$^8$ and 4.6×10$^{11}$ molecules cm$^{-3}$ s after accounting for OH consumption. This suggests that most of the OH that was produced in these OFR369-i(iPrONO/iPrONO-d$_7$) experiments was consumed by α-pinene and its early-generation photooxidation products. We note that OH suppression relative to 254 nm photons, O$_3$, and O is not a concern in OFR369-i(iPrONO), unlike OFR254-iN$_2$O (Peng et al., 2016)."**

R2.8) Abstract line 3: extra "t" before λ.

Deleted (see also reply to R1.3).

R2.9) Equation 1: I assume that density is for the liquid, but please specify.

We modified the text as follows:

P3, L5-6: "where [...] ρ (g cm$^{-3}$) and MW (g mol$^{-1}$) are the organic nitrite **liquid** density and molecular weight…"

R2.10) P1 L17: space needed before (Mao et al., 2009…). This error repeats several times in citations throughout the manuscript.

We fixed this error in the revised manuscript.

R2.11) P2 L13: in the presence of humidified air (if I am understanding the reactions correctly).

No - water vapor is not required for $HO_x$ + $NO_x$ generation via alkyl nitrite photolysis.

R2.12) Page 3, L14: the light manufacturer LCD Lighting is listed in this line but not the previous lines.

Yes, LCD Lighting is the light manufacturer. The part numbers listed in previous lines (F436T5/BL/4P-350, F436T5/BLC/4P-369) were formatted per the preference of LCD Lighting, Inc., where they  manufactured the lamps as OEM equipment and then renamed the end products with part # and reference to Aerodyne Research.

R2.13) P3 L18-23: Is it possible to include some numbers describing this interference (maybe in the SI)? How was the conclusion of "no avail" drawn? Did the 2B monitor read increasing [NOx] with increasing [iPrONO] injection into dark OFR? Since this AN photolysis is a unique aspect of this manuscript, I think instrumental caveats should be better described.

Yes, exactly - the 2B monitor read increasing [$NO_x$] with increasing [iPrONO] injection into dark OFR (both NO and $NO_2$ channels).

We modified the text as follows:

P3, L18-23: "NO and $NO_2$ mixing ratios were measured using a $NO_x$ analyzer (Model 405 nm, 2B Technologies), which quantified [$NO_2$] (ppb) from the measured absorbance at λ= 405 nm, and [NO] (ppb) by reaction with $O_3$ to convert to $NO_2$. Alkyl nitrites introduced to the reactor with the lamps turned off consistently generated signals in the both NO and $NO_2$ measurement channels of the $NO_x$ analyzer, possibly due to impurities **and/or species generated via iPrONO + $O_3$ reactios inside the analyzer**. **For example, background NO and $NO_2$ mixing ratios increased from 0 to 1526 ppb and 0 to 1389 ppb as a function of injected [iPrONO] = 0 to 18.7 ppm** with the lamps off (**Figure S2**). We attempted to correct [NO] and [$NO_2$] for this apparent alkyl nitrite interference by subtracting background signals measured in the presence of alkyl nitrite with lamps off, to no avail, **because background signals (alkyl nitrite present with lamps off) were large compared to signals obtained with alkyl nitrite present with lamps on**. Instead, we constrained [NO] and [$NO_2$] using the photochemical model discussed in Section 2.4."

We added a figure to the supplement (below):

[Figure]

Figure S2. "NO" and "NO$_2$" mixing ratios
measured at the exit of the reactor as a

R2.14): Somewhere in the methods section, the authors should mention what was the flow through the OFR in the calibration experiments. The flow rate through OFR for CIMS experiments is mentioned later, but the flow rate in non-CIMS experiments is not mentioned anywhere.

We modified the text as follows:

P3, L11: "Alkyl nitrites were photolyzed inside a Potential Aerosol Mass (PAM) oxidation flow reactor [...] operated in continuous flow mode (Lambe et al., 2017) **with 5.1±0.3 L/min flow through the reactor unless stated otherwise.**"

R2.15) P8 L5: this sentence is confusing, because it suggests that measured values of NOx are shown in Figure 3, while in fact they are not. Should be reworded accordingly.

The original sentence read: "Figure 3 shows measured and modeled OH$_{exp}$ and NO$_x$ Concentrations". We reworded the sentence to state: "Figure 3 shows measured OH$_{exp}$ and **modeled** NO$_x$ concentrations."

R2.16) P8 L30 (and Figure 4): the explanation of higher NOx offsetting OH production efficiency seems straightforward enough that it should be reproduced by KinSim. However, it seems the model was not run (or not plotted in Figure 4) for this OFR369-i(1,3-Pr(ONO)2) scenario. Can this be explained?

Constraints on the OH rate constant and absorption cross section of 1,3-Pr(ONO)2 are required to model OFR369-i(1,3-Pr(ONO)2). In this case, literature values were not available and we did

not feel we could adequately constrain the rate constant and cross section from first principles or structure-activity relationships.

R2.17): Again, the caption for Figure 4 is confusing because "measured and modeled values … of (iPrONO-d7) and (1,3-Pr(ONO)2)" suggests that the modeled values for BOTH these precursors are plotted, while in fact the model was apparently not run for the latter precursor (this goes back to my previous comment).

We modified the Figure 4 caption as follows:

"$OH_{exp}$ values measured as a function of $I_{369}$ following photolysis of perdeuterated isopropyl nitrite (iPrONO-d7) and 1,3-propyl dinitrite (1,3-Pr(ONO)$_2$). Modeled $OH_{exp}$ values obtained from **OFR369-i(iPrONO-d$_7$)** and OFR369-i(iPrONO) (Fig. 2d) are shown for reference...."

R2.18): Sections 3.3.1 and 3.3.2: are these sub-sections relevant to their parent section 3.3? The parent section title only mentions (iPrONO-d7) and (1,3-Pr(ONO)2). In fact, are these subsections even important enough to be placed in this part of the manuscript? There was no prior discussion of why MeONO and HFiPrONO are important OH precursors. These sub-sections abruptly build up the importance of these two precursors, and then rapidly declare that they are not suitable precursors in the OFR. The narrative flows smoother going directly from experimentally measuring OHexp to setting up estimation equations i.e., from P8 L33 directly to P9 L27. I suggest moving 3.3.1 and 3.3.2 to the end of the manuscript or to the SI.

We moved Section 3.3.1 (MeONO), Section 3.3.2 (now HONO, per reply to R2.1), and Section 3.3.3 (HFiPrONO) to a new Section 3.5 titled: "Anticipated performance of alternative high-NO$_x$ HO$_x$ precursors in OFRs"

R2.19): Sections 3.2 and 3.3 are really hitting the same hammer (how much OHexp is generated from precursor X) on different nails (X = iPrONO, deuterated iPrONO, etc.). I don't see why they need to be separate sections.

We prefer to maintain separate sections for discussion of iPrONO, which presumably will be more widely used, and synthesized alkyl nitrites, which we assume will be used by advanced users. We instead combined the current Sections 3.1 and 3.2 into a single section 3.1 titled "OH$_{exp}$ and NO$_x$ generated from iPrONO photolysis" with subsections 3.1.1 "Effect of photolysis wavelength" and 3.1.2 "Effect of alkyl nitrite concentration".

R2.20): Figure S7b is missing a 1:1 line.

We added the 1:1 line.

R2.21): P10 L18: NO2 needs a subscript.

We added the subscript.

R2.22): P11 L5: OFR operation details (flow rate, etc.) should be described in the Section 2.3. Also, amount of α-pinene injected into OFR should be mentioned to give a sense of the OHR.

We moved some content from P11, L5 to Section 2.3, which now reads as follows:

"In a separate set of experiments, mass spectra of gas-phase α-pinene photooxidation products were obtained with an Aerodyne high-resolution time-of-flight chemical ionization mass spectrometer (Bertram et al., 2011) using nitrate as the reagent ion ($NO_3^-$-HRToF-CIMS, hereafter abbreviated as $NO_3^-$-CIMS) (Eisele and Tanner, 1993; Ehn et al., 2012). [...] The $NO_3^-$-CIMS sampled the reactor output at 10.5 L $min^{-1}$. α-Pinene oxidation products were detected as adducts ions of $NO_3^-$. **In these experiments, the reactor was operated with a residence time of approximately 80 sec to accommodate the undiluted $NO_3^-$-CIMS inlet flow requirement. OFR369-i(iPrONO) and OFR369-i(iPrONO-$d_7$) were operated using $I_{369}$= 6.5×$10^{15}$ photons $cm^{-2}$ $s^{-1}$ and >7 ppm alkyl nitrite. In these experiments, α-pinene was evaporated into the carrier gas by flowing 1 sccm $N_2$ through a bubbler containing liquid α-pinene. Assuming the $N_2$ was saturated with α-pinene vapor, we estimate ~500 ppb α-pinene was introduced to the OFR based on its vapor pressure at room temperature and known dilution ratio into the main carrier gas. In a separate experiment, OFR254-i$N_2$O was operated using $I_{254}$= 3.2×$10^{15}$ photons $cm^{-2}$ $s^{-1}$ + 5 ppm $O_3$ + 1% $H_2O$ + 3.2% $N_2O$. Here, α-pinene was introduced by flowing 1 sccm $N_2$ of a gas mixture containing 150 ppm α-pinene in nitrogen (unavailable for the iPrONO photolysis experiments) into the main carrier gas.**".

R2.23): P11 L11: compound nomenclature is missing some subscripts.

We added missing subscripts to "[($NO_3$)$C_7H_9NO_8^-$]" and "[($NO_3$)$C_7H_{11}NO_8^-$]"

R2.24): P11 L20: this is a cool finding but does not readily jump out in Figure 5. I suggest adding a fourth panel showing a difference between the 5b and 5c (or 5a) spectra and zooming in the m/z scale to show the just a few –OD containing sticks (e.g., from m/z 310 to 360).

We implemented the reviewer's suggestion (please see R2.6).

R2.25) Figure 5: There is enough empty space in each subfigure to include the dinitrite:nitrite ratio value. I suggest adding this in to quantify the "highest ratios observed in 5b" statement on P11 L24.

The revised figure has 6 panels and consequently less empty space to include the dinitrate:nitrate ratio. However, we modified the text to include the dinitrate fractions:

P11, L24: "Second, $C_{10}$ dinitrates were present in all three spectra, with the highest dinitrate:nitrate **fractions** observed in Figures 5b **(0.090)** and 5c **(0.081) and the lowest dinitrate:nitrate fraction observed in Figure 5a** (**0.056**).

R2.26) Figure S7: units of OHexp are incorrect on both X- and Y-axes (s, not s-1).

We changed the units of $OH_{exp}$ from molec $cm^{-3}$ $s^{-1}$ to molec $cm^{-3}$ s.

---

## Author Comment (AC3) · 15 Dec 2018

**Response to reviewers for the paper "HO$_x$ and NO$_x$ production in oxidation flow reactors via photolysis of isopropyl nitrite, isopropyl nitrite-d$_7$, and 1,3-propyl dinitrite at λ = 254, 350, and 369 nm."**

We thank the reviewers for their comments on our paper. To guide the review process we have copied the reviewer comments in black text. Our responses are in regular blue font. We have responded to all the referee comments and made alterations to our paper (**in bold text**).

**Anonymous Referee #3**

**Summary and Recommendation**

Authors introduce a new method for investigating NOx-dependent SOA formation pathways in oxidation flow reactors (OFRs). The new method uses alkyl nitrite photolysis to generate OH and NO2 with two different lights (254 nm and 369 nm). It is an improvement over previous methods used to study NOx-dependent SOA formation pathways in OFRs for three primary reasons. First, because it does not require extremely high levels of ozone and it does not produce nitrate radical as a by-product; both ozone and nitrate radical also contribute to oxidation of SOA precursors and their presence in the reactor creates major challenges for deconvolving contributions from the different oxidants. Second, it can be run with 369 nm lamps and avoid photolytic losses of SOA precursors that can occur with the more commonly used 254 nm lamps. Third, unlike batch reaction chamber studies which can only probe over timescales of hours to ~1 day, the OFR can be used to probe oxidative aging equivalent to multiple days. There were a number of challenges using the new method. First, the alkyl nitrites presented an interference in the NOx analyzer, and they had to use a photochemical model to estimate NOx in lieu of a direct measurement. Second, the NOx generated from alkyl nitrite photolysis introduced an interference in the SO2 analyzer and made it difficult to determine OH exposure. They corrected for this by performing an offline calibration relating SO2 decay and particulate sulfate to OH exposure.The technique does not achieve equivalent OH exposures longer than one day, but does appear to be a promising technique for OFR users to study high NOx SOA chemistry. This is particularly true for oxidation conditions using 369 nm lights. I recommend the manuscript for publication after the following comments are addressed.

R3.1) Elaborate on alkyl nitrite interference with NOx analyzer. Text says they attempted to correct for the interference "to no avail" (p.3, l. 22). What was the issue that prevented this correction?

Please see our response to similar R2.13.

R3.2) OH exposure calibration: not entirely clear how this calibration provided a measure of equivalent OH exposure from OH and NO2 generated from alkyl nitrites. It could be cleared up by better describing the link between Figure S4 and Figure¶ S5. The connection is lost by lack of clarity regarding the x-axis in Figure S4 and explicitly stating at the end of the paragraph how the relationship in Figure S5 is used to estimate OH exposure as presented in Figure S4. In Figure

S4, does the x-axis "sulfate" refers to both SO2 decay in the gas-phase and sulfate measured in the particles with the ACSM? There is something missing in the description that connects how the researchers propose to then presumably use the ACSM sulfate measurement in the presence of alkyl nitrites to estimate initial SO2 (Figure S5) and then relate that back to OH exposure using the relationship shown in Figure S4.

Please see our response to similar R2.3b, including the revised Figure S4 which we are hopeful will provide the necessary clarification to the question raised by Reviewer 3 here.

R3.3) Section 2.4: Additional reactions included in the model and all input parameters, rate constants, etc. are stated very clearly. Thank you. Uncertainties for actinic flux and organic nitrite concentration were also discussed very clearly. This model is being used to estimate NO and NO2 because of the NOx analyzer interference from alkyl nitrites (Section 2.2). I did not see an estimate of the uncertainty for NO and NO2 estimates. Please add those.

Section 2.4 listed estimates of uncertainties for the model inputs: pressure, temperature, [iPrONO], mean residence time, actinic flux, and absorption cross sections and bimolecular rate constants. Because the NO and $NO_2$ mixing ratios are model outputs, the propagated model uncertainties in model input parameters that influence [NO] and [$NO_2$] for the specific model scenarios are represented by the shaded regions in Figures 2 and 3.

R3.4) Figure 2: The explanation for lower OHexp values with 254 nm lights versus 350 nm and 369 nm was not very clear (page 7-8). In particular, elaborate on the concept, "Because σiPrONO;369 « σiPrONO;254 (Table 1), the effect of photolysis wavelength on [NO2] is proportional to σiPrONO, as expected" and how that relates to reduced OH exposure at 254 nm. The other explanation for reduced OHexp with 254 nm lights (decomposition of iC3H7O radical) is described in adequate detail, but it would help to better clarify the proportion of iPrONO that decomposes at this wavelength to provide more context for how significant this pathway is at the shorter wavelengths.

We attempted to clarify this point by modifying the text as shown in our response to reviewer comment R2.4.

R3.5) Section 3.4: OHexp and NO2 estimation equations If this section is going to be included in the main text of the results, the Figure S7 should also be included in the main text since that figure summarizes the main results from this section. Alternatively, the entire section could be moved to supplement. Can you clarify the significance of the results from this section somewhere in the text?

We moved Figure S7 to the main text of the revised manuscript. We modified the text as follows:

P9, L28: "Previous studies reported empirical OH$_{exp}$ algebraic estimation equations for OFR185 and OFR254 (Li et al., 2015; Peng et al., 2015). **These equations parameterize OH$_{exp}$ as a function of readily-measured experimental conditions, therefore providing a simpler**

**alternative to detailed photochemical models that** aid**s** in experimental planning and analysis."

R3.6) One of the goals was to identify the optimal range of conditions for using the new alkyl nitrite method. Can you state the recommended "optimal" conditions more clearly AND also put the OFR conditions within that range into context relative to atmospheric conditions, particularly for NO:NO2 ratios and RO2:HOx? Are there certain conditions that should particularly be avoided? Can you make those more clear as well?

We modified the text as follows:

P12, L16: "Here, we adapted alkyl nitrite photolysis for new OFR applications by characterizing the photolysis wavelength, nitrite concentration, and nitrite composition that result in optimal HOx and NOx generation capabilities. **Based on our results, we recommend photolysis of 5-10 ppm alkyl nitrite at $\lambda \sim$ 365-370 nm photolysis wavelength and >$10^{15}$ photons cm$^{-2}$ s$^{-1}$ actinic flux. If the user has the resources to synthesize iPrONO-d$_7$, better performance is expected relative to iPrONO. Alkyl nitrite photolysis at $\lambda$ = 254 nm is not recommended.** Taken together, OFR254/185-iN$_2$O and OFR369-i(iPrONO/iPrONO-d$_7$) are complementary methods that provide additional flexibility for NOx-dependent OFR studies. OFR369-i(iPrONO)/OFR369-i(iPrONO-d$_7$) generate high-NO$_x$ photooxidation conditions (NO:HO$_2$ ≈10-10000; **NO:NO$_2$ ≈ 0.2-0.7)."**

R3.7) One of the stated benefits of OFRs in the intro is their ability to simulate "multiple days of equivalent atmospheric exposure." (P. 2, L. 4). Looking at Figures 2-4, it does not appear the OFR was capable of obtaining more than 1 day equivalent OH exposure either using the alkyl nitrite technique. Can you comment on how this timescale compares with smog chamber experiments investigating similar NOx pathways?

We note that the text on P2, L4 (which is part of the introduction) refers to other versions of the OFR chemistry described in previous publications.

Otherwise, one of the goals of this work was to achieve HOx generation via photolysis of 1,1,1,3,3,3-hexafluoroisopropyl nitrite (HFiPrONO) because modeling suggested it is capable of simulating multiple days of OH exposure due to its extremely low OH reactivity. This detail was omitted from the discussions manuscript but we have added it to the revised manuscript to provide additional context. As described in the paper, we were unable to synthesize HFiPrONO, following literature methods.

We modified the text as follows:

P9, L15: "We **predict** that OFR369-i(HFiPrONO) should attain higher OH$_{exp}$ than OFR369-i(iPrONO) and OFR369-i(iPrONO-d$_7$) due to similar photolysis rates (Andersen et al., 2003) and ~200 times lower OH reactivity of HFiPrONO/hexafluoroacetone relative to iPrONO/acetone (Atkinson et al., 1992; Tokuhashi et al., 1999). **Simple modeling calculations suggest that application of OFR369-i(HFiPrONO) may achieve up to a week of equivalent OH exposure."**

P12, L21: "We anticipate that alkyl nitrite photolysis is suitable for the characterization of first-generation, high-NOx photooxidation products of most precursors, **at OH$_{exp}$ comparable to environmental chambers investigating high-NO$_x$ conditions.**"

R3.8) It would be helpful to see a direct comparison between OH exposure using the alkyl nitrite method versus other techniques that have been used to probe the high NOx pathway (for example, N2O addition).

We modified the text as shown in our response to the similar reviewer comment R1.1.

R3.9)P. 2, L. 28-30: alkyl nitrite were stored in amber vials and refrigerated until use. Can you clarify how long they can be stored and approximately how much time passed from synthesis to experimental use in these experiments?

We consulted with Pfaltz and Bauer, Inc., the company from which we obtained isopropyl nitrite. We received the following email response from a sales manager on 29 October 2018:

"Andy,

Per your request for the shelf life of our I10550, 2 years is a general guide.

We have seen 3 years if refrigerated and the container kept tightly closed away from moisture.

Regards,

Bob Milburn
Inside Sales Manager
Pfaltz & Bauer, Inc.
172 E. Aurora St.
Waterbury, CT 06708
tel  203-574-0075  X102
fax 203-574-3181
bobm@pfaltzandbauer.com
www.pfaltzandbauer.com"

We modified the text as follows:

"The resulting clear yellow liquid was dried over sodium sulfate, neutralized with excess sodium bicarbonate, and then stored in amber vials and refrigerated at 4°C until use **(within one week of synthesis in this work)**. **Under these storage conditions, the nominal shelf life of isopropyl nitrite and similar organic nitrites is approximately 2 years (B. Milburn, personal communication, 29 October 2018).**"